# Sororin is an evolutionary conserved antagonist of WAPL

Ignacio Prusén Mota [1,2,3], Marta Galova[4], Alexander Schleiffer [4], Tan-Trung Nguyen [1,2], Ines Kovacikova[2], Carolina Farias Saad[1,2,3], Gabriele Litos[4], Tomoko Nishiyama[4], Juraj Gregan [2,5] ✉, Jan-Michael Peters [4] ✉ & Peter Schlögelhofer [1,2] ✉

Cohesin mediates sister chromatid cohesion to enable chromosome segregation and DNA damage repair. To perform these functions, cohesin needs to be protected from WAPL, which otherwise releases cohesin from DNA. It has been proposed that cohesin is protected from WAPL by SORORIN. However, in vivo evidence for this antagonism is missing and SORORIN is only known to exist in vertebrates and insects. It is therefore unknown how important and widespread SORORIN's functions are. Here we report the identification of SORORIN orthologs in *Schizosaccharomyces pombe* (Sor1) and *Arabidopsis thaliana* (AtSORORIN). *sor1Δ* mutants display cohesion defects, which are partially alleviated by *wpl1Δ*. *Atsororin* mutant plants display dwarfism, tissue specific cohesion defects and chromosome mis-segregation. Furthermore, *Atsororin* mutant plants are sterile and separate sister chromatids prematurely at anaphase I. The somatic, but not the meiotic deficiencies can be alleviated by loss of WAPL. These results provide in vivo evidence for SORORIN antagonizing WAPL, reveal that SORORIN is present in organisms beyond the animal kingdom and indicate that it has acquired tissue specific functions in plants.

Eukaryotic cells perform a complex series of events in order to equally distribute the replicated genome among their daughter cells. DNA replication is not immediately followed by karyokinesis and the newly formed sister chromatids are physically linked for long periods of time until their disjunction during mitosis or meiosis[1,2]. Sister chromatid cohesion (SCC) is mediated by the cohesin complex, which is thought to topologically entrap DNA helices from both newly replicated sisters[3,4]. While SCC promotes chromosome biorientation and DNA damage repair, cohesin can also extrude loops of DNA and facilitate distant intra-chromatid interactions, supporting further roles in chromatin organization and gene expression[5].

Cohesin's core subunits have been identified and characterized in all branches of the eukaryotic kingdom including yeast and plants[6,7]. As a member of the Structural Maintenance of Chromosome (SMC) protein family, cohesin is formed by a heterodimer of SMC1 and SMC3. These proteins fold back on themselves at the hinge domain, where they interact with each other, to form long antiparallel coiled-coil structures. At the other end, their ATPase head domains are bridged together by an α-kleisin subunit, RAD21 (also known as Scc1 or Mcd1) or its meiotic counterparts REC8 and RAD21L[8–10]. These heterotrimeric ring-like structures crucially depend on the recruitment of SCC3 (SA or STAG proteins) to fulfil their chromatin-related functions. SCC3 contributes to cohesin loading, maintenance on chromosomes and its subsequent release from DNA[11–16]. Together, these four proteins form the cohesin core complex.

[1]Max Perutz Labs, Vienna Biocenter Campus (VBC), Vienna, Austria. [2]University of Vienna, Center for Molecular Biology, Department of Chromosome Biology, Vienna, Austria. [3]Vienna Biocenter PhD Program, a Doctoral School of the University of Vienna and the Medical University of Vienna, Vienna, Austria. [4]Research Institute of Molecular Pathology (IMP), Vienna Biocenter (VBC), Vienna, Austria. [5]Department of Applied Genetics and Cell Biology, Institute of Microbial Genetics, University of Natural Resources and Life Sciences, Tulln an der Donau, Austria. ✉e-mail: juraj.gregan@univie.ac.at; Jan-Michael.Peters@imp.ac.at; peter.schloegelhofer@univie.ac.at

In addition to SCC3, two further HAWK proteins (HEAT repeat proteins Associated With Kleisin), SCC2 (also known as NIPBL or Mis4) and PDS5[17], bind to kleisin in a mutually exclusive manner to regulate cohesin behaviour[18,19]. SCC2 is needed to stimulate cohesin's ATPase activity[16,18,20,21] and has been proposed to load cohesin onto DNA[11,22]. In vitro experiments have shown that NIPBL is further required for cohesin-mediated loop extrusion[20,21]. PDS5 and WAPL can disrupt the interaction between the SMC3 and kleisin subunits, thereby releasing cohesin from chromatin[23–26]. While cohesin shows a highly dynamic behaviour through cycles of association and release from chromatin, especially during G1, a fraction of cohesin becomes stably bound to DNA after replication and mediates SCC[27,28]. Establishment of cohesion during DNA replication requires acetylation of two lysine residues on SMC3 by the conserved acetyltransferase Eco1/CTF7[29–33]. In yeast, Pds5 is required for the acetylation process and for stabilizing cohesin on chromatin[25]. Inactivation of cohesin loading during G1 induces complete cohesin dissociation from DNA in a Wpl1-dependent manner, whereas if inactivation takes place during G2, some cohesin remains chromatin-bound[28]. In *A. thaliana*, mutation of four of the five *PDS5* genes leads to mild defects in meiosis and to severe deficiencies in development, fertility and somatic homologous recombination (HR)[34]. Inactivation of both copies of WAPL in *A. thaliana* only mildly affects overall plant development and fertility[35], but rescues the dramatic somatic deficiencies associated with loss of *CTF7*[36,37].

In vertebrates and *Drosophila*, an additional protein factor, Sororin, is recruited to the cohesin complex in a replication and SMC3-acetylation-dependent manner[38–43]. Sororin promotes SCC until the onset of anaphase by displacing WAPL from PDS5 and counteracting its releasing effects[40]. Both WAPL and Sororin bind to PDS5 through conserved FGF and YSR motifs[40,44].

In somatic cells, Sororin accumulates on chromatin between S and G2 phases and becomes dispersed in the cytoplasm after nuclear envelope breakdown except at centromeric regions where it persists until metaphase[40,42], consistent with its function in promoting SCC[43,45]. This suggests that Sororin, as the cohesin complex, is removed from chromosomes in a stepwise manner[46]. First, the so-called prophase pathway removes chromosomal arm cohesin in a non-proteolytic manner during the first stages of mitosis and meiosis. This process largely depends on WAPL and phosphorylation of STAG2[13,23,47,48]. Sororin phosphorylation has been proposed to participate in both processes: Cdk1-phosphorylated Sororin may act as a docking protein and recruit Polo-like kinase 1 (Plk1) to mediate STAG2 phosphorylation[49]. Besides, Aurora B and Cdk1 phosphorylate Sororin on several sites and destabilise its association with PDS5, thereby promoting WAPL-mediated removal of cohesion[50,51]. At centromeres, the Shugoshin-PP2A complex protects cohesin from the prophase pathway by keeping Sororin and cohesin subunits in a dephosphorylated state[51–53]. During the metaphase-to-anaphase transition, the anaphase-promoting complex/cyclosome (APC/C[Cdc20]) targets phosphorylated Securin for degradation to promote the separase-mediated cleavage of the phosphorylated kleisin subunit[42,54–56].

Current data suggest that the main function of Sororin is to counteract the activity of WAPL. While WAPL appears conserved across kingdoms, including yeasts and land plants, no conserved WAPL antagonist has been described so far. SMC3 acetylation has been proposed to be sufficient to counteract the function of WAPL in organisms thought to lack Sororin, like yeast and plants[37,57]. In *Drosophila melanogaster*, the Sororin-related protein Dalmatian has been characterized[40]. Dalmatian combines protein functions of Sororin and Shugoshin to promote and protect cohesion[41]. Recently, a meiosis I-specific WAPL antagonist (SWI1), that shares no sequence homology to Sororin, has been characterized in *A. thaliana*[58].

To identify possible homologues of *Sororin* we performed a thorough bioinformatics analysis. Our searches revealed putative Sororin relatives in various lower and higher eukaryotes. Here we show that *S. pombe* Sor1 is required for efficient sister chromatid cohesion and that *wpl1* deletion partially suppresses defects caused by the *sor1Δ* mutation. We also demonstrate that Sor1 physically interacts with the cohesin subunit Psm3 (SMC3) and Pds5. We furthermore show, that the *A. thaliana* Sororin homologue (*AtSORORIN*) is essential for vegetative development and microsporogenesis. Lack of AtSORORIN leads to tissue-specific reduction or loss of SCC and chromosomal mis-segregation. Consistent with AtSORORIN's proposed function, these somatic phenotypes can be alleviated by loss of WAPL. *Atsororin* mutant plants are sterile, affected in male meiosis with chromatids displaying premature loss of cohesion and splitting of sister-centromeres at anaphase I. Interestingly, the meiotic defects cannot be alleviated by loss of WAPL. Taken together, we provide the first organismal in vivo evidence for Sororin antagonizing WAPL function and demonstrate that Sororin is an evolutionary conserved cohesin regulator that has acquired additional functions in plants.

## Results

### *S. pombe* Sor1 and *A. thaliana* AtSORORIN share sequence similarities with metazoan Sororin proteins

To identify possible orthologs of *Sororin*, we performed a comprehensive bioinformatics analysis using sensitive remote homology searches. Our searches revealed putative Sororin proteins in both lower and higher eukaryotes including various yeast and plant species. They all show only weak overall sequence conservation with their vertebrate counterparts but they share various characteristic features. The *S. pombe* (SPAC9E9.05) and the *A. thaliana* (At3g56250) gene candidates which both encode short proteins were analysed in detail. Vertebrate Sororin and Wapl proteins interact with Pds5 through their YSR and FGF motifs[40,44]. Whereas SPAC9E9.05 has a putative FGF motif, such sequence is not present in the plant candidate. A KEN box targets vertebrate Sororin and *Drosophila* Dalmatian for APC/C[Cdh1]-dependent degradation, but has not been found in either the plant (At3g56250) or the yeast (SPAC9E9.05) candidates. Similar to metazoan Sororin, the proteins encoded by SPAC9E9.05 and At3g56250 have a conserved motif, referred to as the Sororin domain, preceded by a K/R-rich domain at their C-termini (Fig. 1a). The Sororin domain has been implicated in interactions with STAG2 and contains two conserved phenylalanine residues important for the maintenance of sister chromatid cohesion (Fig. 1b)[59,60].

The *S. pombe* Sororin candidate, SPAC9E9.05, has so far been annotated as a poorly characterized *Schizosaccharomyces* specific protein[61]. Interestingly, a *SPAC9E9.05* deletion mutant was identified in a screen for mutants that showed negative synthetic growth interaction with the cohesion-defective mutants *eso1-G799D* (Eso1 is the *S. pombe* ortholog of Esco1/2 and CTF7[62]) and *mis4-242* (Mis4 is the *S. pombe* ortholog of NIPBL[63]), suggesting that *SPAC9E9.05* may be involved in regulation of sister chromatid cohesion[64]. Given the similarity of *S. pombe* SPAC9E9.05 and *Arabidopsis* At3G56250 with metazoan Sororin and the data presented below, we decided to name their encoding genes *sor1* (*Sor*orin-like *1*) and *AtSORORIN*, respectively. Using this extended information we compiled Sororin-like sequences and further cohesion regulators (WAPL[23] and Haspin[65]) from evolutionary distant species to illustrate that each has a common ancestor and was lost in some lineages during evolution (Supplementary Fig. 1).

### *S. pombe* Sor1 is a nuclear protein involved in sister chromatid cohesion

If *S. pombe* Sor1 was functionally related to mammalian Sororin, then it should be present in the nucleus. Nuclear localization of Sor1 was previously observed when expressed under the control of a strong *nmt1* promoter[66]. To analyse Sor1 localization, we expressed Sor1-GFP from its native promoter. In an asynchronously growing culture, Sor1-GFP localized to the nucleus in most cells (Supplementary Fig. 2a). Immunostaining experiments confirmed the nuclear localization of

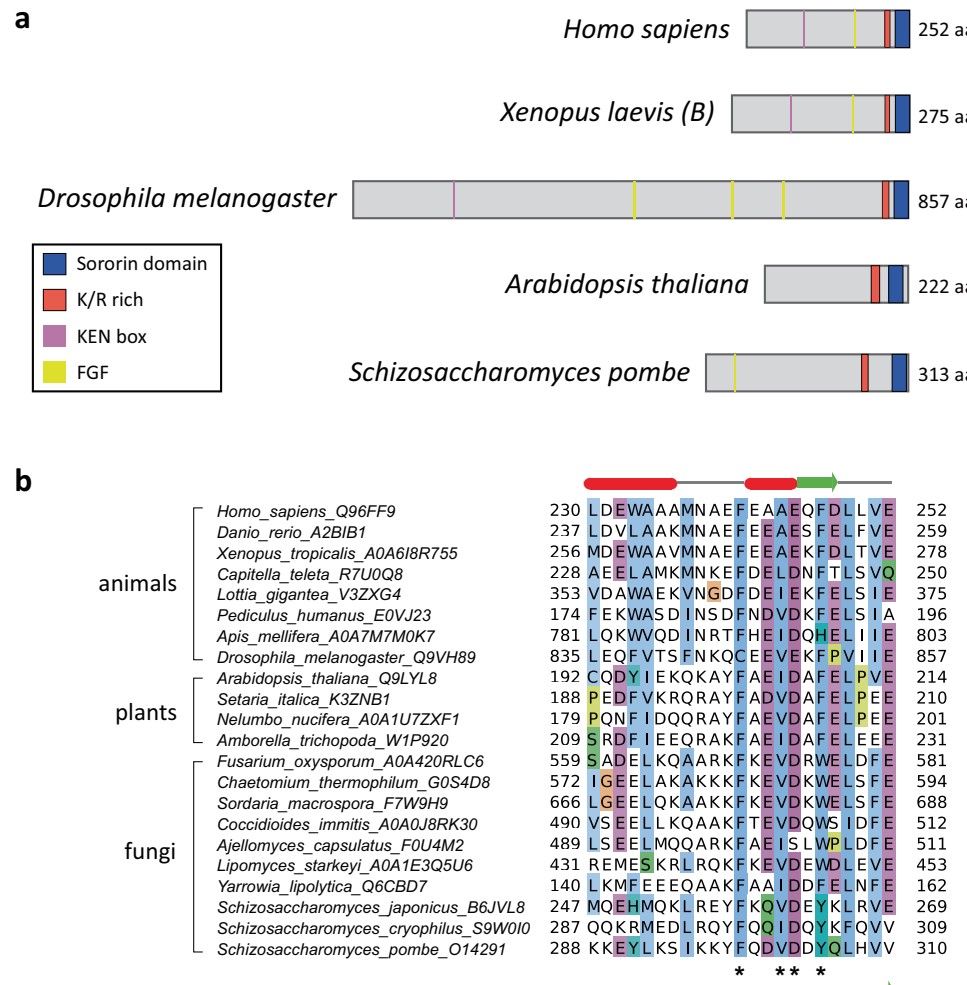

**Fig. 1 | *S. pombe* Sor1 and *A. thaliana* AtSORORIN share sequence similarities with metazoan Sororin proteins.** **a** Domain architecture of Sororin and putative Sororin orthologs. The Sororin domain is shown in blue, the cluster of positively charged residues (lysine, arginine) in red, the KEN box in magenta and the FGF motif in yellow. The domain graphs were created with the help of the domain illustrator (DOG 2.0[112]). **b** Alignment of the C-terminal Sororin domain. UniProt accessions are provided next to the species names. Residues mutated in this study are indicated by asterisks. Secondary structure prediction of *H. sapiens* and *S. pombe* are shown on the top and bottom, respectively, where alpha helices are in red, and beta strands in green.

Sor1-Flag during all tested cell cycle stages (Supplementary Fig. 2b). Western blot analysis showed that Sor1-Pk protein levels are lower in nitrogen starved/G1 cells as compared to cycling cells which are mostly in G2 (Supplementary Fig. 2c).

To assess the role of Sor1 in regulation of cohesion, we analysed sister chromatid cohesion at the centromeric region (*cen2*-GFP) of chromosome 2. In metaphase, *sor1Δ* mutant cells showed a small, but significant, increase of split sister centromeres (Fig. 2a), indicative of a cohesion defect between sister centromeres. However, the role of Sor1 in sister chromatid cohesion is not essential because we observed no defects in chromosome segregation in *sor1Δ* cells (Fig. 2b). In mammalian cells, Sororin is dispensable for sister chromatid cohesion in the absence of WAPL[40]. We therefore analysed sister chromatid cohesion in cells lacking Wpl1, the fission yeast ortholog of WAPL[62]. Interestingly, the increase in split sister centromeres in *sor1Δ* mutant cells was prevented in *sor1Δ wpl1Δ* double mutants (compared to wild type), suggesting that similarly to mammalian cells *wpl1* deletion reduces the sister chromatid cohesion defect caused by the *sor1Δ* mutation (Fig. 2a).

Deletion of *sor1* showed negative synthetic growth interaction with both *eso1-G799D* and *mis4-242* mutations but the cause of these defects is unknown[64]. We asked whether defective segregation of chromosomes contributes to this growth defect. Indeed, we observed a higher frequency of lagging chromosomes associated with a higher rate of chromosome mis-segregation in *eso1-G799D sor1Δ* and *mis4-242 sor1Δ* double mutants as compared to single mutants (Fig. 2b). This observation is consistent with the role of Sor1 in sister chromatid cohesion regulation.

In telophase and G1, mammalian Sororin is targeted by APC/C for degradation[42]. We therefore tested whether the fission yeast Sor1 is an APC/C substrate, despite the lack of a defined KEN box. We added in vitro translated Sor1-HA to interphase *Xenopus* egg extracts in the presence of cycloheximide followed by addition of Cdh1 to activate APC/C^Cdh1. We also added in vitro translated Sor1-HA to meiotic metaphase-arrested CSF extracts in the presence of cycloheximide followed by addition of CaCl$_2$ to activate APC/C^Cdc20. As expected, activation of APC/C^Cdc20 led to a rapid degradation of the APC/C^Cdc20 substrate Cyclin B2 and also endogenous *Xenopus* Sororin was degraded within few minutes after activation of APC/C^Cdh1. However, we did not observe degradation of *S. pombe* Sor1 by either APC/C^Cdh1 or APC/C^Cdc20 (Supplementary Fig. 2d).

**Conserved residues in the Sororin domain are important for Sor1 function and association with cohesin**

Mammalian Sororin physically interacts with cohesin and Pds5 and these interactions are essential for Sororin's function[40,59,60]. If the fission yeast Sor1 was an ortholog of metazoan Sororin, Sor1 should

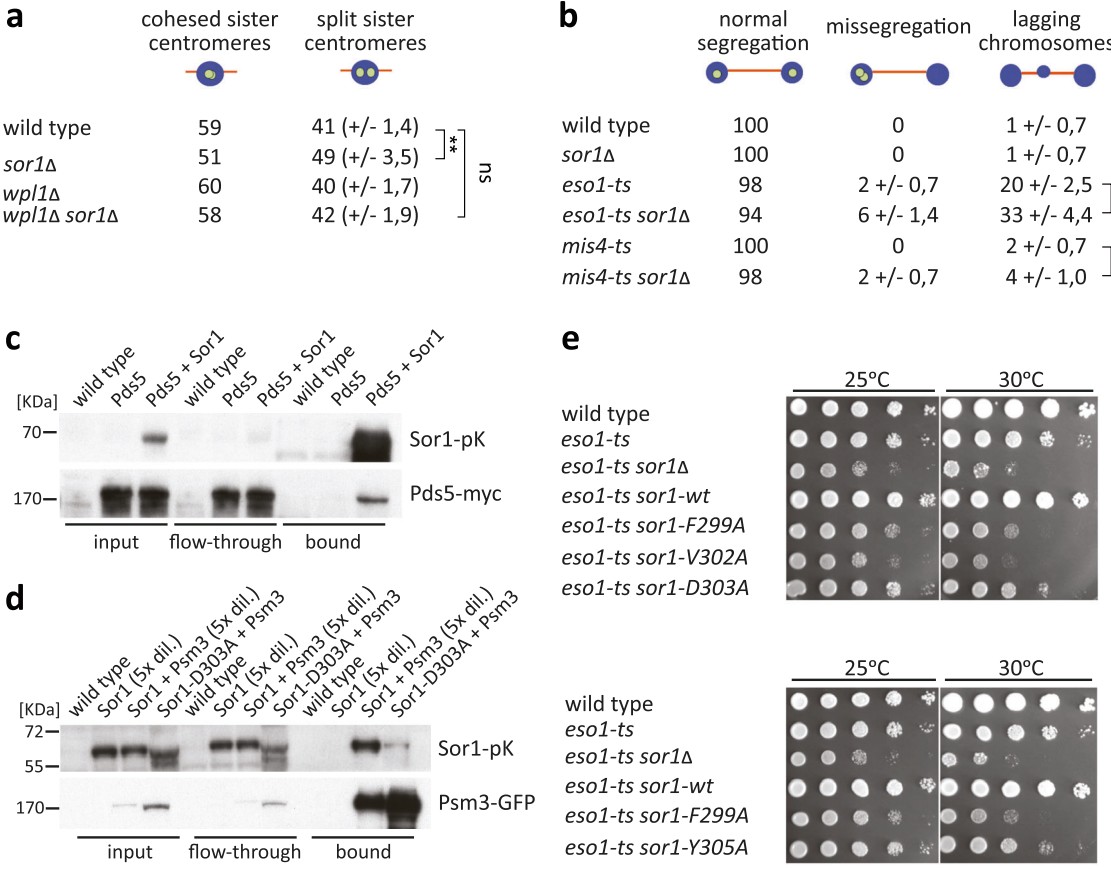

**Fig. 2 | S. pombe Sor1 is involved in sister chromatid cohesion and its conserved residues are important for Sor1 function and association with cohesin. a** *sor1Δ* cells show a weak cohesion defect which is partially suppressed by *wpl1Δ*. Wild type and *sor1Δ* haploid cells expressing *cen2*-GFP were fixed and stained with antibodies against tubulin and GFP and sister chromatid cohesion was analysed in metaphase cells. Nuclei were visualized by Hoechst staining. Means +/− standard deviations are shown. Unpaired t-test was performed for statistical analysis (**p < 0.01; ns−not significant). **b** Negative synthetic growth interaction in *eso1tssor1Δ* and *mis4tssor1Δ* double mutants are associated with chromosome segregation defects. Wild type, *sor1Δ*, *eso1-G799D (eso1-ts)*, *eso1-G799D sor1Δ*, *mis4-242 (mis4-ts)* and *mis4-242 sor1Δ* haploid cells expressing *cen2*-GFP were fixed and stained with antibodies against tubulin and GFP. Nuclei were visualized by Hoechst staining. Samples were examined under the fluorescence microscope, and segregation of chromosome 2 marked by *cen2*-GFP was scored in late anaphase cells. Lagging chromosomes were identified as Hoechst-staining bodies between the poles of the spindle in late anaphase. Means +/− standard deviations are shown. Unpaired t-test was performed for statistical analysis (*p < 0.05; **p < 0.01). **c** Pds5-Myc co-immunoprecipitates with Sor1-Pk. Protein extracts were prepared from cycling wild-type cells and cells expressing Sor1-Pk, Pds5-myc or both Sor1-Pk and Pds5-Myc, as indicated. Proteins bound to anti-V5 agarose beads, which bind the Pk tag on Sor1, were analysed for Pds5-Myc by Western blotting using anti-Myc antibody. The experiment was repeated twice with similar results. **d** Psm3-GFP co-immunoprecipitates with Sor1-Pk and this interaction is weakened by mutating conserved Sor1 residue D303. Protein extracts were prepared from cycling wild-type cells and cells expressing Sor1-Pk, Psm3-GFP or both Sor1-Pk and Psm3-GFP, as indicated. Proteins bound to anti-GFP agarose beads, which bind the GFP tag on Psm3, were analysed for Sor1-Pk by Western blotting using anti-V5 antibody. Mutant protein Sor1-D303A-Pk co-immunoprecipitated with the Psm3-GFP protein less efficiently, as compared to wild-type Sor1-Pk. The experiment was repeated twice with similar results. **e** The four conserved residues in the Sororin domain are important for the Sor1 function. Strains with the indicated mutations were grown on YES medium for one day. Serial dilutions were spotted onto YES plates and incubated for 3 days at 25 °C or 30 °C. While expression of a wild type Sor1 rescued the growth defect of the *eso1-G799D sor1Δ* double mutant (*eso1-ts sor1-wt*), mutant Sor1 proteins carrying F299A, V302A, D303A or Y305A substitutions did not rescue the growth defect of *eso1-G799D sor1Δ* double mutants (*eso1-ts sor1-F299A, eso1-ts sor1-V302A, eso1-ts sor1-D303A, eso1-ts sor1-Y305A*).

interact with cohesin and/or Pds5. We indeed observed that Pds5-Myc co-immunoprecipitated with Sor1-Pk and Sor1-Pk co-immunoprecipitated with Psm3-GFP (Fig. 2c, d).

The Sororin domain is required for sister chromatid cohesion and association of Sororin with cohesin in mammalian cells[59,60]. To test whether the Sororin domain of *S. pombe* Sor1 is important for its association with cohesin in fission yeast, we analysed the ability of Psm3-GFP to immunoprecipitate mutant protein Sor1-D303A-Pk, in which a conserved aspartic acid residue D303 in the Sororin domain has been replaced by alanine. Sor1-D303A-Pk co-immunoprecipitated less efficiently with the Psm3-GFP protein, compared to wild-type Sor1-Pk, suggesting that the conserved residue D303 in the Sororin domain of Sor1 is important for the association of Sor1 with cohesin (Fig. 2d).

We then asked whether the interaction between *S. pombe* Sor1 and cohesin is functionally relevant. As expected, expression of a wild-type

Sor1 rescued the growth defect of the *eso1-G799D sor1Δ* double mutant to the level of the *eso1-G799D* single mutant. However, expression of the Sor1-D303A mutant, which weakens the interaction between Sor1 and cohesin, did not restore the growth defect of *eso1-G799D sor1Δ* double mutants (Fig. 2e). Mutating three other conserved residues in the Sororin domain of Sor1 (F299A, V302A and Y305A) resulted in a similar phenotype (Fig. 2e). The observed mutant phenotype was not due to lack of Sor1 expression as all four Sor1 mutant proteins (Sor1-D303A-TAP, Sor1-F299A-TAP, Sor1-V302A-TAP and Sor1-Y305A-TAP) were expressed, although at reduced levels (Supplementary Fig. 2e).

Taken together, we show that fission yeast Sor1 shares similarity with metazoan Sororin proteins. Sor1 is associated with the cohesin complex and *sor1Δ* mutant cells show defects consistent with the role of Sor1 in regulation of sister chromatid cohesion. Conserved residues at the C-terminus of Sor1 are important for the Sor1 function and its

association with cohesin. Unlike metazoan Sororin proteins, Sor1 is not essential for sister chromatid cohesion, suggesting that fission yeast possesses mechanisms that are able to compensate for the absence of Sor1. Our results are consistent with the notion that Sor1 is an ortholog of Sororin in the fission yeast *S. pombe*.

### A. thaliana SORORIN is essential for vegetative development and microsporogenesis

Our findings obtained in *S. pombe* motivated us to analyse a Sororin candidate in a non-vertebrate higher eukaryote. The *A. thaliana SORORIN* gene candidate (At3g56250) consists of four exons and codes for a relatively small protein (222 amino acids). Using CRISPR-Cas9 technology we generated a 5 bp deletion in its first exon, creating a premature stop codon (Fig. 3a). The deletion was confirmed by sequencing the genomic region and also the mRNA transcripts (converted to cDNA) of *Atsororin −/−* plants (Supplementary Fig. 3a). According to the nature of the CRISPR-Cas9 mediated short deletion, the *AtSORORIN* gene expression in wild-type and *Atsororin −/−* mutant plants was not significantly different in all analysed tissues (Supplementary Fig. 3b). Heterozygous *Atsororin +/−* plants appear like wild type with only minimally reduced seed numbers, but homozygous mutants display a prominent dwarf phenotype, have few and short siliques and epinastic rosette leaves with short petioles that grow around an undersized stem (Fig. 3b). This dramatic phenotype can be complemented with a transgene containing the wild-type gene, including all up- and down-stream regulatory sequences and introns (Supplementary Fig. 3c, d), corroborating that the mutation in the *AtSORORIN* gene indeed caused the observed aberrations. Plant roots and shoots develop from meristems, which are formed by actively dividing cells that self-renew and differentiate into new tissue. Root development is severely affected by the lack of AtSORORIN. *Atsororin* mutant plant roots grow significantly shorter than those of wild type, and they completely lose the characteristic layered cellular organization (Fig. 3c, d). Moreover, mutant plants are sterile since their short siliques do not develop viable seeds (Fig. 3e).

Heterozygous, self-pollinated *Atsororin +/−* plants have less than 4% homozygous *Atsororin −/−* offspring, representing a significant deviation from the expected Mendelian segregation ratio (Fig. 3f). Reciprocal crosses between *Atsororin +/−* heterozygous mutant plants and wild type plants revealed that the distortion of segregation ratios is exclusively caused by the male generative cells (Fig. 3g). In fact, the morphology of *Atsororin −/−* mutant anthers is abnormal, their size decreased and the amount of shed pollen strongly reduced. A test for pollen viability (Alexander staining) showed that unlike wild-type plants, *Atsororin −/−* mutants produce only very few pollen grains of which only very few are viable (Fig. 3h). In *Atsororin +/−* heterozygous mutant plants the regularly sized anthers develop a variable number of non-viable pollen grains. We assume that most pollen grains lacking AtSORORIN die prior to fertilization, in line with the transmission distortion reported above (Supplementary Fig. 3e).

### Loss of WAPL rescues Atsororin-associated defects

In mammalian cells, Sororin is needed to counteract the cohesin-releasing activity of Wapl, and therefore deficiencies related to loss of Sororin can be suppressed by loss of Wapl[40]. Arabidopsis *wapl1-1 wapl2* double mutants exhibit normal vegetative growth and only a mild reduction in fertility (Fig. 3b)[35]. The *Atsororin*-associated somatic defects can be suppressed by the *wapl1-1 wapl2* double mutant, underlining that Arabidopsis *SORORIN* is a bona fide relative of its vertebrate counterpart. In the *Atsororin wapl1-1 wapl2* triple mutant normal growth of the aerial plant parts and of the roots is restored (Fig. 3b–d).

WAPL inactivation only leads to a limited rescue of the fertility defect observed in *Atsororin* mutants. *Atsororin wapl1-1 wapl2* anthers are nearly as small as those of *Atsororin* single mutants and only very

few viable pollen grains are formed (Fig. 3h). Correspondingly, the triple mutant produces only very few seeds, but still significantly more than the *Atsororin* single mutant (wild type $55 \pm 4$ seeds/silique ($n = 74$), *wapl1-1 wapl2* $36 \pm 9$ seeds/silique ($n = 144$; $p < 0.0001$), *Atsororin* $0.096 \pm 0.35$ seeds/silique ($n = 52$; $p < 0.0001$); *Atsororin wapl1-1 wapl2* $5 \pm 4$ seeds/silique ($n = 166$; $p < 0.0001$) (Fig. 3e).

### AtSORORIN is essential in a sub-set of tissues

The data, especially the epistatic relation to *WAPL*, suggested that the gene product of *AtSORORIN* acts in a similar manner as its vertebrate counterpart. We anticipated that the most obvious molecular phenotype of *Atsororin* mutants should be pre-mature loss of sister-chromatid cohesion. To analyse chromosome numbers and sister chromatid cohesion we prepared mitotic cell nuclei samples and specifically stained centromeres (via fluorescent in situ hybridization, FISH).

Indeed, interphase nuclei from roots of *Atsororin* mutant plants contain on average $16.82 \pm 3.68$ centromere signals ($n = 34$). This is significantly more compared to wild type ($10.02 \pm 0.1458$ centromere signals; $n = 93$; $p < 0.0001$), *wapl1-1 wapl2* double mutants ($10.32 \pm 1.66$ centromere signals; $n = 73$; $p < 0.0001$) and *Atsororin wapl1-1 wapl2* triple mutants ($10.49 \pm 1.83$ centromere signals; $n = 59$; $p < 0.0001$) (Fig. 4a, b). We attribute the severe mis-organisation of cells in the *Atsororin* mutant roots (Fig. 3c) (Supplementary movies 1–4) and the arbitrary chromosome numbers in interphase nuclei to massive chromosome mis-segregation due to pre-mature loss of cohesin. Since the homozygous *Atsororin* mutant plants are under-represented and the mutant root material is scarce and experimentally difficult to process we could only obtain a few cells at metaphase. While in wild-type 10 doublet signals can be seen, in *Atsororin* plants individual chromatids are arranged at the metaphase plate. The anticipated pre-mature loss of SCC leads to random segregation of chromatids during anaphase in *Atsororin* mutants. Importantly, *Atsororin wapl1-1 wapl2* triple mutants are much less affected than the *Atsororin* single mutant (Figs. 3c and 4a, b).

Somatic interphase cell nuclei isolated from leaves of *Atsororin* mutant plants, had a close to regular number of chromosomes ($10.3 \pm 0.5746$ centromere signals; $n = 53$), which is still significantly different when compared to wild-type plants (10 centromere signals; $n = 84$; $p < 0.0001$), *wapl1-1 wapl2* double mutants (10 centromere signals; $n = 68$; $p < 0.0001$) or *Atsororin wapl1-1 wapl2* triple mutants (10 centromere signals; $n = 82$; $p < 0.0001$) (Supplementary Fig. 3f, g).

We also prepared somatic cells from inflorescences, containing a large number of actively dividing cells that can be readily processed and analysed (Fig. 4c). As for the leaf cells, we established first the number centromeric signals of interphase nuclei. We found, similar to the numbers obtained from leaf cells and in contrast to the ones obtained from root cells, that most cells contain the correct number of chromosomes in *Atsororin* mutants ($10.26 \pm 1.25$ centromere signals; $n = 266$) but still significantly different when compared to wild type (10 centromere signals; $n = 224$; $p < 0.0001$), *wapl1-1 wapl2* double mutants ($10.01$ centromere $\pm 0.09$ centromere signals; $n = 238$; $p < 0.0001$) or *Atsororin wapl1-1 wapl2* triple mutants ($10.09$ centromere signals; $n = 236$; $p < 0.0001$) (Fig. 4d).

Since a large number of actively dividing cells in anaphase could be observed in the inflorescence tissue, we were also in the position to monitor chromosome segregation. In accordance with the mild aberrations of chromosome numbers in interface nuclei, we observed mostly regular chromosome disjunction in *Atsororin* nuclei from inflorescences (97% symmetric disjunction, $n = 133$) with 10 separating chromosomes at either side of the division plane. Those *Atsororin* plants (13/133) that carried 11 chromosomes in all cells, most likely obtained via a gamete with a supernumerary chromosome, showed regular disjunction. In this sense, the occurrence of symmetric divisions was not significantly different from wild-type plants ($n = 112$,

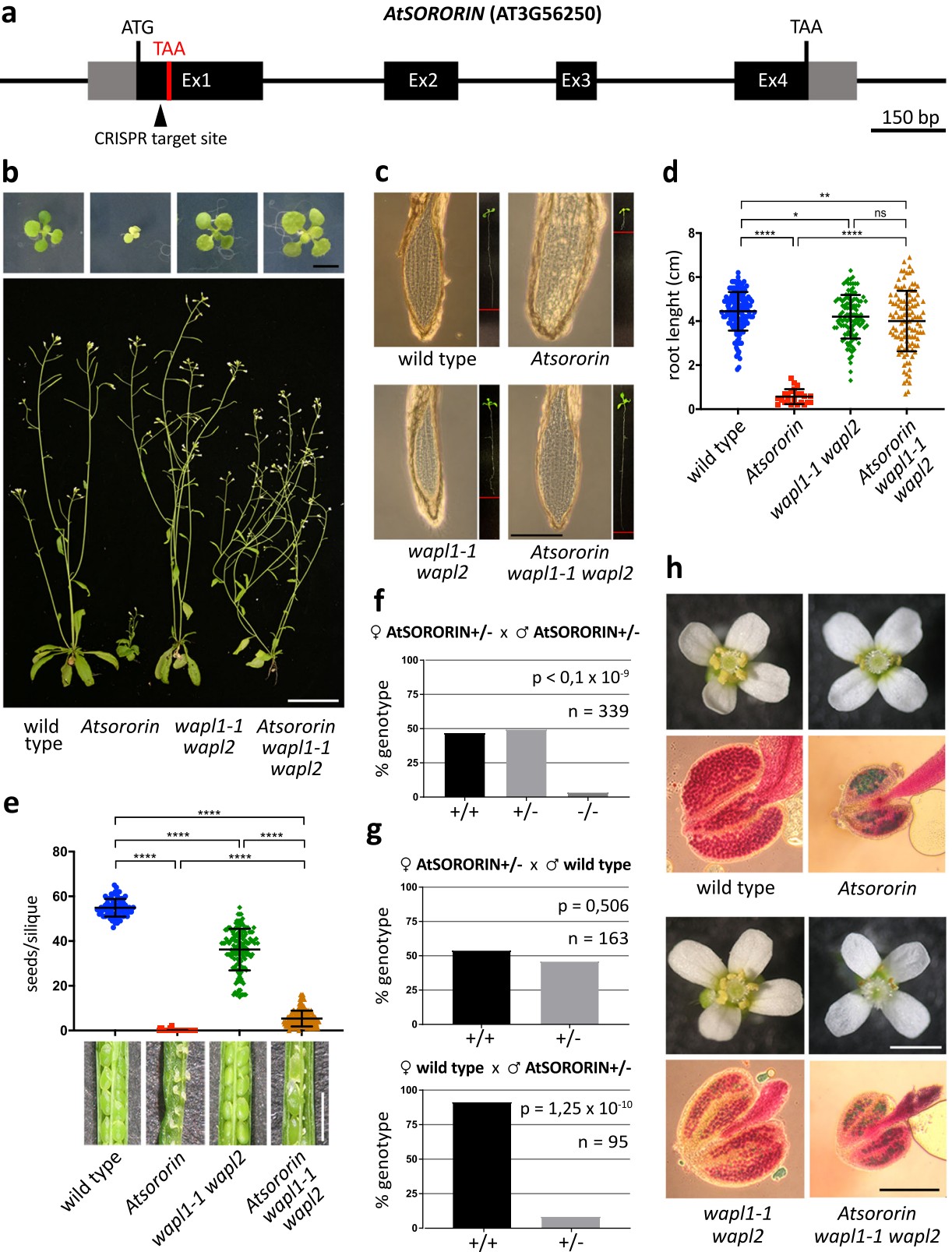

*p* = 0.2525) and *wapl1-2 wapl2* (*n* = 119, *p* = 0.9999) and *Atsororin wapl1-2 wapl2* (*n* = 118, *p* = 0.6246) mutants. Moreover, staining for SCC3 in somatic cells of inflorescences of wild-type and *Atsororin* mutants showed no difference in SCC3 abundance on chromosomes during prophase and interphase, indicating normal distribution and abundance of cohesin in these cells in *Atsororin* mutants (Supplementary Fig. 4a).

We also measured the inter-sister centromere distance during prophase and prometaphase (Fig. 4c, e, f) in somatic cells from inflorescences. Post S-phase 10 doublet signals can be seen in all genotypes tested. While the mutation in *AtSORORIN* does not lead to complete loss of cohesion between sister chromatids, the distance between the 10 centromeric doublet signals is significantly increased compared to wild type (Prophase: 457.6 nm in *Atsororin*, *n* = 39;

**Fig. 3 | Loss of WAPL rescues somatic defects of *Atsororin* mutants. a** Schematic representation of *AtSORORIN* (AT3G56250) gene, with 5′ and 3′ UTRs (grey boxes), introns (black lines) and exons (black boxes), open reading frame (ATG/TAA, black), Cas9 target site (black triangle) and premature stop codon in mutant plans (TAA, red) indicated. **b** The severe growth restriction of homozygous *Atsororin* mutants plants (seedlings, scale bar = 5 mm; mature plants, scale bar = 5 cm) is alleviated by loss of WAPL (*Atsororin wapl1-1 wapl2* triple mutants). Wild-type plants, *Atsororin*, *wapl1-1 wapl2* double mutants and *Atsororin wapl1-1 wapl2* triple mutants were grown side-by-side for comparison. **c** Images of root tips and entire seedlings (small pictures) of plants grown on media plates for two weeks. Root growth restriction (red bars) and loss of characteristic layering of the root meristem in *Atsororin* mutant plants are evident. These deficiencies are rescued by loss of WAPL (*wapl1-1 wapl2* double mutants). All plants were grown side-by-side for comparison. Scale bar = 1 mm. **d** Quantification of root growth of wild-type plants (*n* = 139), *Atsororin* −/− plants (*n* = 23), *wapl1-1 wapl2* plants (*n* = 113) and *Atsororin wapl1-1 wapl2* plants (*n* = 113) grown on media plates for two weeks. Error bars represent standard deviations of the mean. Unpaired Mann-Whitney test has been applied (\**p* < 0.05; \*\**p* < 0.01; \*\*\*\**p* < 0.0001; ns−difference not significant). **e** Loss of fertility in *Atsororin* mutant plants is only partially rescued by WAPL inactivation.

Wild type plants (*n* = 74), *Atsororin* −/− plants (*n* = 52), *wapl1-1 wapl2* plants (*n* = 144) and *Atsororin wapl1-1 wapl2* plants (*n* = 166) were grown side-by-side and genotypes are indicated. Error bars represent standard deviations of the mean. Unpaired Mann-Whitney test has been applied (\*\*\*\**p* < 0.0001). Images show representative, opened siliques and developing seeds. *Atsororin wapl1-1 wapl2* triple mutant plants have siliques with some seeds, which are mostly bigger than those formed in wild-type plants. Scale bar = 1 mm. **f** Genotypes of offspring of self-pollinated *AtSORORIN* +/− plants. The homozygous *Atsororin* −/− genotype is strongly underrepresented (chi-square analysis, *p* value indicated). **g** Genotyping the offspring of reciprocal crosses between *AtSORORIN* +/− and wild-type plants indicates that only male, but not female, gametogenesis, is affected by the *Atsororin* mutation (two-sided Fisher's exact test, *p* values indicated). **h** Flower architecture is not affected by the lack of AtSORORIN whereas anther growth and pollen viability are severely disturbed. *Atsororin* single mutants (*n* = 20) and *Atsororin wapl1-1 wapl2* triple mutants (*n* = 17) develop smaller anthers with few viable pollen grains than wild type (*n* = 8) and *wapl1-1 wapl2* mutants (*n* = 12). All plants were grown side-by-side and genotypes are indicated. The term "*Atsororin* mutant" refers to the homozygous mutant allele configuration if not explicitly stated otherwise. Scale bar flowers = 1 mm, scale bar anthers = 200 μm.

378.9 nm in wild type, *n* = 45; *p* < 0.0001. Prometaphase: 699 nm in *Atsororin*, *n* = 55; 562.9 nm in wild type, *n* = 42; *p* < 0.0001). The sister-centromere distance is, as anticipated, significantly shortened in *wapl1-2 wapl2* mutants compared to wild type (Prophase: 294.2 nm in *wapl1-2 wapl2*, *n* = 37; *p* < 0.0001. Prometaphase: 481.4 nm in *wapl1-2 wapl2*, *n* = 46; *p* < 0.01). During prophase, the *Atsororin wapl1-2 wapl2* triple mutants have a centromeric distance that is not different from wild type (349.2 nm, *n* = 50; *p* = 0.2695), significantly shorter than the *Atsororin* single mutant (457.6 nm, *p* < 0.0001) and increased when compared to *wapl1-2 wapl2* mutants (294.2 nm, *p* < 0.0001). At prometaphase the centromeric distance of *Atsororin wapl1-2 wapl2* is as tight as in the *wapl1-2 wapl2* mutant (446.1 nm in *Atsororin wapl1-2 wapl2*, *n* = 49; *p* = 0.4004). It is interesting to note that in *wapl1-2 wapl2* double mutants, cohesion of sister chromatid arms is maintained in prometaphase since no individual arms can be distinguished. This also holds true in the *Atsororin wapl1-2 wapl2* triple mutant background.

Taken together, we conclude that both AtSORORIN and WAPL impact sister-chromatid cohesion, and that AtSORORIN is not the exclusive antagonist of WAPL activity in all somatic plant tissues.

## AtSORORIN is needed for centromeric sister chromatid cohesion during male meiosis

Our analysis indicated that somatic divisions in root cells and microsporogenesis are most severely affected by loss of *AtSORORIN*. To analyse if the underlying cause for the latter can be related to a perturbation of male meiosis we prepared chromosome spreads from meiocytes. Comparing wild-type and *Atsororin* meiocytes it is apparent that AtSORORIN is not an essential factor for sister chromatid cohesion in prophase I. Meiocytes from *Atsororin* plants show normal chromosome condensation and pairing during pachytene and also chiasmata at diakinesis. Bivalents were properly orientated at the metaphase I plate. Yet, in anaphase I sister chromatids split prematurely and were subsequently segregated at random in meiosis II (Fig. 5). While in anaphase I/telophase I we observed 5 DAPI-stained bodies at each pole of the dyad in wild type, in *Atsoronin* mutants around 10 DAPI stained bodies can be seen. In metaphase II these 10 DAPI stained bodies could not be aligned properly, were distributed at random during anaphase II and subsequently led to unbalanced tetrads. Supernumerary DAPI stained bodies, which we interpret as individual chromatids, were detected in 71% of *Atsororin* meiocytes during prophase II-metaphase II stages (*n* = 38), while this was never observed in wild type (*n* = 67; *p* < 0.0001).

The *wapl1-2 wapl2* double mutants showed strengthened cohesion, characterized by the distinct shape of bivalents at metaphase I, as previously described[35], and regular distribution of chromosomes at meiosis I (*n* = 32) and II. Importantly, in male meiocytes

of *Atsororin wapl1-2 wapl2* triple mutants, premature loss of sister chromatids persists. Supernumerary chromatids were observed in 80% of all anaphase I/telophase I meiocytes in the triple mutant (*n* = 40; *p* < 0.0001 compared to wild-type or *wapl1-2 wapl2*). This means, that the premature loss of centromeric sister chromatid cohesion at anaphase I/telophase I in *Atsororin* mutants cannot be rescued by loss of WAPL.

To determine the precise timing of loss of sister chromatid cohesion during meiosis of *Atsororin* mutant plants we performed centromeric FISH analysis on meiotic spreads (Fig. 6a). As mentioned above, homologous chromosome pairing appeared normal in *Atsororin* mutants, underlined by the presence of 5 dominant CEN signals observed at pachytene stage. During late metaphase I/early anaphase I, five pairs of CEN signals were observed in wild type, with two distinct signals per bivalent (each signal representing two fused sister centromeres) that were orientated to opposite poles. In *Atsororin* mutants, homologous chromosomes showed proper bipolar orientation at metaphase I but the centromeric signals pointing to either pole were often split. All of the observed *Atsororin* metaphases had more than 10 CEN signals (*n* = 24), indicating that sister chromatid centromeres were not fused as in wild type (Fig. 6a, b). We quantified the number of centromeric signals observed at metaphase I (including cells from metaphase I to prophase II stages) and metaphase II stages in wild-type plants and *Atsororin*, *wapl1-2 wapl2* double and *Atsororin wapl1-2 wapl2* triple mutants (Fig. 6c, d). While meiocytes from wild-type and *wapl1-2 wapl2* mutant plants did mostly not suffer from premature splitting of sister-centromeres at metaphase I (93.4% and 91.3% of cells with 10 centromere signals respectively, *n* = 76 in wild type, *n* = 23 in *wapl1-2 wapl2*; *p* = 0.6622) and had perfectly paired sister-centromeres at metaphase II (*n* = 17 in wild type, *n* = 10 in *wapl1-2 wapl2*), Atsororin and *Atsororin wapl1-2 wapl2* mutants displayed split sister-centromere signals at metaphase I (*n* = 24, *p* < 0.0001 in *Atsororin*; *n* = 31, *p* < 0.0001 in the triple mutant), and non-paired sister-centromeres at metaphase II (*n* = 13, *p* < 0.0001 in *Atsororin*; *n* = 17, *p* < 0.0001 in *Atsororin wapl1-2 wapl2* mutants).

As mentioned above, after loss of sister chromatid cohesion, progression through meiosis II is compromised and in *Atsororin* and *Atsororin wapl1-2 wapl2* mutants individual chromatids segregated at random. We quantified tetrads with balanced chromosome numbers (Fig. 6e). While in wild-type plants, all meiocytes generated balanced tetrades (*n* = 33), none of the *Atsororin* mutants produced balanced tetrades (*n* = 25; *p* < 0.0001), *wapl1-2 wapl2* mutants produced 75% of balanced tetrades (*n* = 28; *p* < 0.01) and *Atsororin wapl1-2 wapl2* none (*n* = 18; *p* < 0.0001). These observations lend further support to the notion that the meiotic deficiencies in AtSORORIN cannot be rescued by loss of WAPL.

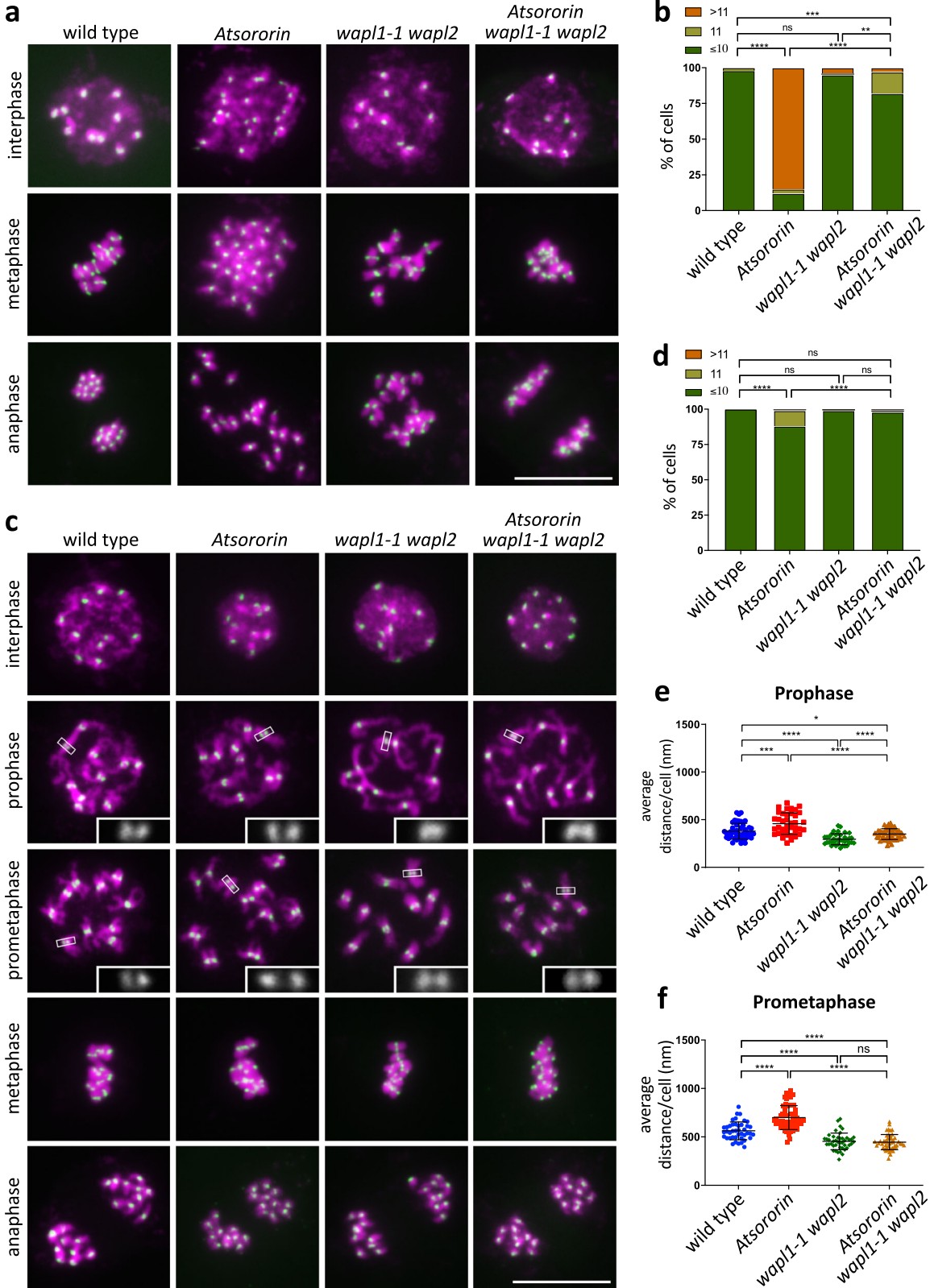

It Is interesting to note that while univalent chromosomes were not observed in WT or *wapl1-1 wapl2* mutants, a significant fraction (13%) of *Atsororin* meiocytes showed presence of an extra univalent chromosome (scored at diakinesis-metaphase I stages; $n = 53$; $p < 0.01$). Presence of extra chromosomes could be the consequence of a previous non-disjunction event in the meiocyte precursor cells, or the result of fertilization between unbalanced generative cells (see also

above). Interestingly, we did not observe univalents in the *Atsororin wapl1-1 wapl2* triple mutants ($n = 32$).

**AtSORORIN does not affect meiotic cohesin abundance and axis formation in meiotic prophase**

We were curious to understand AtSORORIN's impact on cohesion abundance in a severely affected tissue. We therefore performed

**Fig. 4 | Somatic defects in *Atsororin* mutants are tissue-specific and WAPL-dependent.** DNA was stained with DAPI (magenta) and fluorescence in situ hybridization (FISH) was performed to detect centromeric regions (green). **a** Spreads of root cell nuclei. Interphase, metaphase and anaphase stages were analysed for wild-type plants and *Atsororin*, *wapl1-1 wapl2* and *Atsororin wapl1-1 wapl2* mutants. Scale bar = 10 μm. **b** Quantification of centromeric-FISH signals in interphase root nuclei. *Atsororin* mutants ($n = 34$) show a significantly higher number of signals than wild type ($n = 93$), *wapl1-1 wapl2* ($n = 73$) and *Atsororin wapl1-1 wapl2* ($n = 59$) (Fisher's exact test; **$p < 0.01$; ***$p < 0.001$; ****$p < 0.0001$; ns−difference not significant). **c** Spreads of somatic cell nuclei from inflorescences. Interphase, prophase, pro-metaphase, metaphase and anaphase stages were analysed for wild-type plants and *Atsororin*, *wapl1-1 wapl2* and *Atsororin wapl1-1 wapl2* mutants. Magnifications of signals at the sister centromeres are provided for prophase and prometaphase stages. Scale bar = 10 μm. **d** Quantification of centromeric-FISH signals observed in nuclei of cells from inflorescences at interphase. Quantification was performed on wild-type plants ($n = 224$) and *Atsororin* ($n = 266$), *wapl1-1 wapl2* ($n = 238$) and *Atsororin wapl1-1 wapl2* mutants ($n = 236$) (Fisher's exact test; ****$p < 0.0001$; ns−difference not significant). **e** Measurements of the physical distance between FISH signals of sister chromatid centromeres during prophase. *Atsororin* mutants ($n = 39$) show a significant increase in distance between sister centromeres when compared to wild type ($n = 45$), *wapl1-1 wapl2* ($n = 37$) and *Atsororin wapl1-1 wapl2* ($n = 50$). Error bars represent standard deviations of the mean. Unpaired t-test was performed (*$p < 0.05$; ***$p < 0.001$; ****$p < 0.0001$). **f** Measurements of the physical distance between FISH signals at sister chromatid centromeres during prometaphase. *Atsororin* mutants ($n = 55$) show a significant increase in distance between sister centromeres when compared to wild type ($n = 42$), *wapl1-1 wapl2* ($n = 46$) and *Atsororin wapl1-1 wapl2* ($n = 49$). Error bars represent standard deviations of the mean. Unpaired t-test was performed (****$p < 0.0001$; ns−difference not significant).

chromosome spreads of male meiocytes and subsequent immune-staining using antibodies directed against the cohesin subunit SCC3 and the meiosis-specific kleisin subunit REC8 (Fig. 7; Supplementary Fig. 4b). We scored cells at the zygotene/pachytene transition as cohesins can still be observed well at this stage. To correctly stage progression of meiosis, we also detected the meiotic axis component ASY1 and the transverse filament protein of the synaptonemal complex (SC), ZYP1. Our analysis shows that during meiotic prophase, axis formation, as judged from the ASY1 signal, and SC formation, as judged from the ZYP1 signal, is indistinguishable from wild type in *Atsororin*, *wapl1-2 wapl2* and *Atsororin wapl1-2 wapl2* mutants. Furthermore, cohesion abundance and deposition, as judged from the SCC3 and REC8 signals, along the chromosome arms appears unaffected in *Atsororin*, *wapl1-2 wapl2* and *Atsororin wapl1-2 wapl2* mutants (Fig. 7; Supplementary Fig. 4b).

## Discussion

Cohesin complexes are evolutionarily ancient inventions of nature, involved in proper chromosome disjunction in mitosis and meiosis, but also essential for chromosome organization[67]. In animal cells, Wapl has been recognized as a cohesin removal factor which itself is kept in check by the antagonizing protein Sororin[23,40,45]. While cohesion complex proteins, Wapl and Eco1-dependent acetylation of cohesin are conserved from yeast and plants to humans, Sororin was thought to be present only in metazoans[5,41]. A Sororin-like protein has been characterized in the fly, with a peculiar dual function; it serves as a Wapl antagonist and also as a centromeric cohesion protector[40,41]. It has also been suggested that in yeast the conserved acetyltransferase Eco1 takes over the function of Sororin[68]. In *Arabidopsis*, the protein SWI1 antagonizes the function of WAPL, but exclusively only during meiotic prophase I, and it shares no sequence homology with the vertebrate or fly relatives[58,69]. These results suggested that WAPL antagonists should also be present in the genomes of other eukaryotes, but possibly strongly diverged in sequence or occurring as functional domain in the context of larger proteins.

Applying sensitive remote homology searches, we identified putative Sororin relatives in various organisms, including *S. pombe* and *A. thaliana*, which are separated by ~1.5 billion years of independent development. Moreover, collecting sequences of Sororin-like proteins from distant lineages revealed that it is an evolutionary old cohesin regulator, but lost in some taxa.

We show that Sor1, the *S. pombe* Sororin-relative, physically interacts with cohesin (via SMC3/Psm3) and Pds5. However, we observed only a mild sister chromatid cohesion defect in *sor1Δ* cells, suggesting that there are other mechanisms that compensate for the absence of Sor1. Importantly, *wpl1* deletion partially suppressed the sister chromatid cohesion defect caused by the *sor1Δ* mutation, suggesting that, similarly as metazoan Sororin, Sor1 antagonizes the function of Wapl. Our results are consistent with the notion that Sor1 is an ortholog of Sororin in the fission yeast *S. pombe*.

Conversely, the *Arabidopsis* Sororin relative is an important factor for plant viability and vigour. *Atsororin* mutant plants are under-represented in segregating populations due to compromised male, but not female, transmission of the mutant allele. The few plants that develop with a homozygous *Atsororin* mutation are dwarfed, have a short and distorted root and are sterile. Interestingly, among the somatic tissues analysed, only roots show a strong chromosome mis-segregation phenotype, while other tissues are less affected. Somatic cells from inflorescences show hardly any mis-segregation but a widening of centromeric distances in prophase/pro-metaphase, compatible with AtSORORIN's role in limiting WAPL's activity. *Atsororin* plants are sterile and the main underlying cause appears to be premature loss of sister centromere cohesion at anaphase I during male meiosis. This is different from the defect observed in *swi1* mutants, with premature loss of sister chromatid cohesion in early meiotic prophase I[58]. Importantly, the somatic defects of *Atsororin* mutants and the meiotic defect of *swi1* mutants could be rescued in the absence of WAPL (*wapl1 wapl2* double mutants), while the meiotic defects of *Atsororin* could not be alleviated.

We assume that only a sub-fraction of cohesin complexes is regulated by AtSORORIN during plant meiosis. Specifically in meiosis I, SWI1[58] antagonizes WAPL during meiotic prophase I and AtSORORIN possibly only during the metaphase I to anaphase I transition. The latter is possible not apparent in a *swi1* mutant as all cohesive cohesin has been lost during prophase, leading to premature separation of sister chromatids. Importantly, in a *Atsororin* mutant, sister chromatid cohesion is maintained until metaphase I, yet centromeric fusion is lost at anaphase I onset leading to a peculiar co-segregation of separated sister chromatids to the same pole. As only a small fraction of centromeric cohesin complexes seem affected in the *Atsororin* mutant, this may not be sufficient for detecting differences in cytological preparations.

It is interesting to note, that a very similar phenotype compared to *Atsororin* has been observed in the acetyltransferase mutant *CTF7*[36], a relative of Eco1 and ESCO1/2[31,70]. Eco1/CTF7 acetylates the cohesin subunit SMC3 during DNA replication, thereby promoting recruitment of SORORIN and antagonizing the function of WAPL[38–40]. In plants, inactivation of WAPL in a *ctf7* mutant background restores somatic growth but fails to fully rescue the *ctf7* fertility defect[37]. These results indicate that first, AtSORORIN and AtCTF7 may act in the same pathway to promote sister chromatid cohesion by antagonizing WAPL, and second, that the dramatic dwarf phenotype observed in the single *Atsororin* and *ctf7* mutants is not a direct effect of the respective mutation, but an indirect, possibly mediated by altered cohesin dynamics. In the future, we will investigate the precise epistatic and functional relation of CTF7, SWI1 and AtSORORIN.

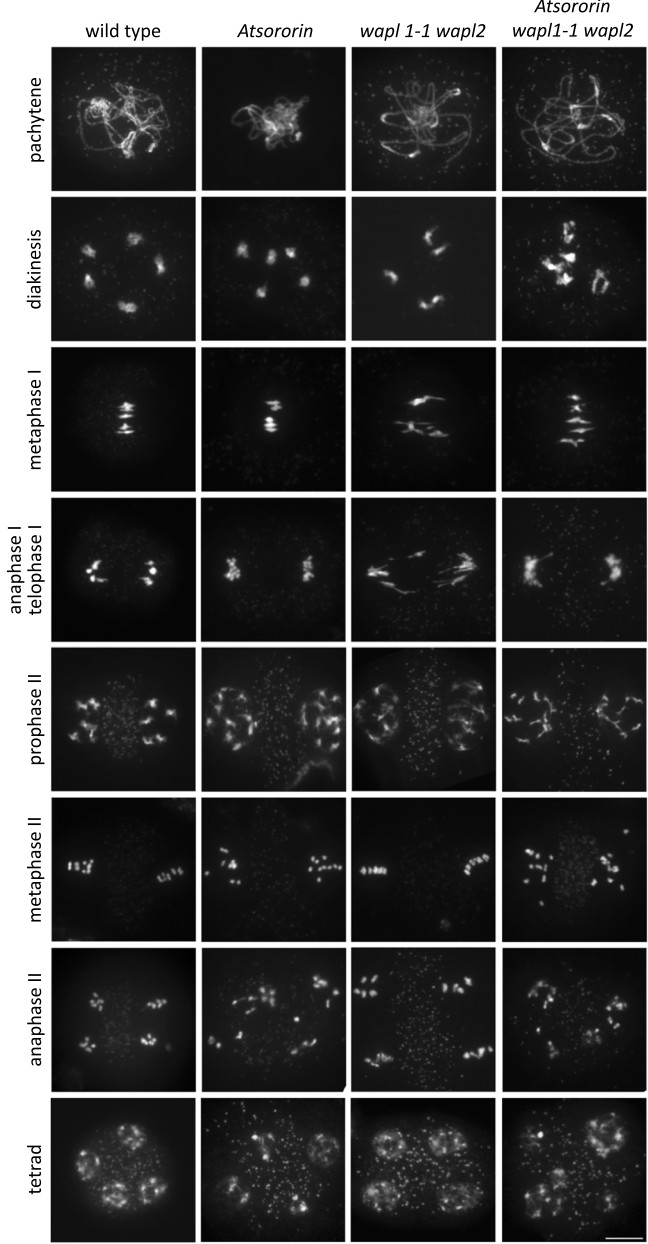

wild type *Atsororin* *wapl 1-1 wapl2* *Atsororin wapl1-1 wapl2*

pachytene

diakinesis

metaphase I

anaphase I / telophase I

prophase II

metaphase II

anaphase II

tetrad

**Fig. 5 | Plants lacking AtSORORIN exhibit defects during male meiosis.** Spreads of meiotic nuclei from wild-type plants and *Atsororin*, *wapl1-1 wapl2* and *Atsororin wapl1-1 wapl2* mutants. Meiotic progression until metaphase I, including homologous chromosome pairing and bivalent formation, appears normal in all genotypes. The number of DAPI-stained bodies is increased in mutants lacking AtSORORIN after metaphase I, yielding 10 chromatids in prophase II. Progression through meiosis II is therefore defective in *Atsororin* single mutants with the subsequent formation of unbalanced tetrads. Inactivation of WAPL does not rescue chromosome non-disjunction observed in anaphase II in the *Atsororin* single mutants. For each stage and genotype at least 5 pictures, all showing similar results, were acquired. Scale bar = 10 μm.

Sororin has initially been perceived as the only WAPL antagonist in vertebrates[40], but later the histone kinase Haspin has also been described as a WAPL antagonist with respect to cohesive cohesin[65,71]. It is interesting to note that loop extruding cohesin is also protected from WAPL by CTCF[72,73]. Haspin has been implicated in centromeric localization of the chromosome passenger complex (CPC) which plays a crucial role in chromosome bi-orientation by correcting erroneous microtubule attachment[74]. Localization of the CPC relies on histone H3-T3 phosphorylation, which is mediated by the histone kinase

Haspin/Hrk1[75–77]. Hrk1/Haspin localization to centromeres depends on its interaction with Pds5[65,78,79].

In this sense, the protein PDS5 has emerged as a central regulator for the orchestration of cohesin dynamics. Via its conserved A P D/E A P motif[44,79], it can interact with diverse regulators. In human cells, PDS5 utilises this motif to interact with WAPL, HASPIN and SORORIN. Importantly, the three proteins share a common PDS5-interaction motif (PIM: K/R T/S Y S R K/L) and compete for PDS5 binding[44,65,71]. Furthermore, *S. pombe* Pds5 has been characterized to interact with Wpl1, Hrk1 and Eso1 (with the latter two inhibiting cohesin removal)[79]. Also these three proteins have a common Pds5-interaction motif[79] and compete for the same binding domain on Pds5. Here we demonstrate that yet another protein, Sor1, can interact with Pds5, potentially also competing for the same binding platform.

*Arabidopsis* has five *PDS5* genes[34], of which three encode PDS5 variants with a perfectly conserved interaction motif. The two *A. thaliana* WAPL proteins have well-conserved PIMs at their N-termini (R T Y G R R) and are very likely direct interaction partners of PDS5 proteins, with experimental proof for the WAPL1-PDS5A pair[58].

Common to all SORORIN proteins is the Sororin domain[60]. Previously it was shown to be important for interaction with cohesin complexes (SA2) and the maintenance of sister chromatid cohesion[59,60]. The Sororin domain is well-conserved in the *A. thaliana* and in *S. pombe* relatives, yet only one phenylalanine is present within the motif of the latter. We established in *S. pombe* that mutating this residue (F299) to alanine is as detrimental as a complete deletion of the *sor1* gene.

Interestingly, while we could not identify a putative PIM in the *A. thaliana* Haspin protein we noticed a well-conserved Sororin domain (Y F R D I D A F E), which is not present in Haspin proteins from other organisms. In this sense, plant Haspin may be localised to cohesin via interacting with the SCC3 subunit and may also play a role as WAPL antagonist in plants.

Importantly, our study provides the first organismal in vivo evidence that SORORIN antagonizes WAPL. We conclude (1) that orthologs of SORORIN are wide-spread in eukaryotes including yeast and plant species; (2) that plants encode more than one WAPL antagonist, and (3) that they act in clearly defined tissue and developmental contexts; and (4) that AtSORORIN may have acquired, similar to *Drosophila*'s Dalmatian, additional WAPL-independent functions in sister centromere protection at the meiosis I to meiosis II transition.

## Methods
### Bioinformatic analyses
Sororin orthologs are characterized by a very short domain at the C-terminus, which is shared between mammals and insects. This region consists of a stretch of positively charged amino acids, a polar linker (varies in size between 10 and 20 amino acids) and a conserved motif predicted to form two alpha helices and a beta strand[40]. We could not expand the Sororin protein family to other taxonomic clades such as fungi or plants when we considered only statistically significant hits (e-value 1e−2). To identify candidates in other model organisms we used a hidden Markov model (HMM) of the C-terminal region (covering the *Homo sapiens* Sororin protein gi|18087845|ref|NP_542399.1|: 216-252) and searched specifically in the proteomes of *Saccharomyces cerevisiae* and *Schizosaccharomyces pombe* (HMMER suite version 2.3.2)[80]. We received 26 (*S. cerevisiae*) and 28 (*S. pombe*) hits with low significant e-values between 0.78 and 10. The hits were manually filtered according to the following criteria: location of the alignment at the C-terminus, conservation of the hydrophobic pattern (especially the phenylalanine residues), and no overlap with known functional domains. In budding yeast, no hit fulfilled all these criteria. In fission yeast, the best hit was to the protein SPAC9E9.05.1 (e-value 1, score −4.0). The protein is 313 residues long and the HMM alignment spanned from 241 to 310. SPAC9E9.05.1 is specific to the

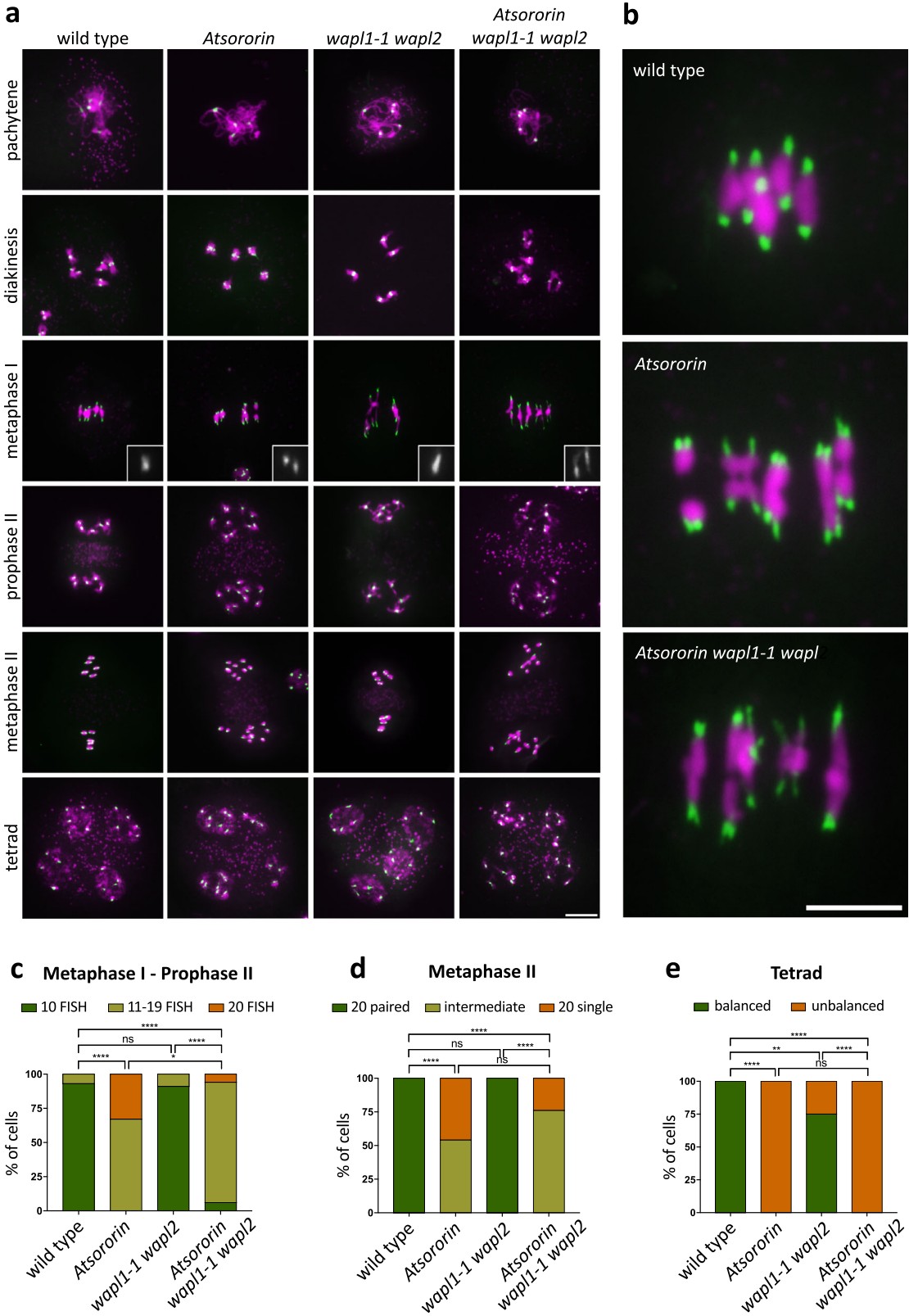

*Schizosaccharomyces* genus—no other orthologs could be detected with a NCBI-blastp search (version 2.2.26)[81] besides in *Schizosaccharomyces cryophilus*, *Schizosaccharomyces octosporus*, and *Schizosaccharomyces japonicus*. The conservation within the SPAC9E9.05 protein family is very poor (overall *S. pombe* and *S. japonicus* are only 23% identical), the C-terminus being the highest conserved region (30% identical). No known functional domains could be detected in the PFAM database. We incorporated the SPAC9E9.05.1 *Schizosaccharomyces* sequences into the HMM model and extended the search to other fungi species. In the proteome of the ascomycete *Pyrenophora tritici-repentis* (strain Pt-1C-BFP), the best hit was to a predicted protein (gi|189210197|ref|XP_001941430.1|, score 13.5, e-value 0.089) belonging to an uncharacterized protein family that is conserved within the *Pezizomycotina clade*.

**Fig. 6 | Premature separation of centromeres during meiosis in *Atsororin* mutant plants.** Fluorescence in situ hybridization experiment on male meiocytes with a probe directed against the centromeric regions (green) in wild-type plants and *Atsororin*, *wapl1-1 wapl2* and *Atsororin wapl1-1 wapl2* mutants. **a** Inactivation of *AtSORORIN* leads to premature loss of centromeric cohesion at the metaphase to anaphase transition during meiosis I. Inlays show magnifications of sister centromeric signals during metaphase I. Scale bar = 10 μm. **b** Magnifications of images depicting metaphase I stages for wild-type plans, *Atsororin* single mutants and *Atsororin wapl1-1 wapl2* triple mutants. Premature splitting of centromeric signals is evident in the absence of AtSORORIN. Centromeres are stained in green and DNA is stained in magenta. Scale bar = 10 μm. **c** Quantification of the number of centromeric signals from metaphase I to prophase II stages in wild-type plants ($n = 73$) and *Atsororin* ($n = 25$), *wapl1-1 wapl2* ($n = 22$) and *Atsororin wapl1-1 wapl2* ($n = 33$) mutants. Fischer's exact test was performed (*$p < 0.05$; ****$p < 0.0001$; ns−difference not significant). **d** Quantification of the number of centromeric signals at metaphase II in wild-type plants ($n = 13$) and *Atsororin* ($n = 12$), *wapl1-1 wapl2* ($n = 12$) and *Atsororin wapl1-1 wapl2* ($n = 15$) mutants. Fischer's exact test was performed (****$p < 0.0001$; ns−difference not significant). **e** Quantification of the number of centromeric signals in tetrads (balanced: 4 nuclei with 5 centromere signals each) in wild type plants ($n = 33$) and *Atsororin* ($n = 25$), *wapl1-1 wapl2* ($n = 28$) and *Atsororin wapl1-1 wapl2* ($n = 18$) mutants. Two-sided Fischer's exact test was performed (**$p < 0.01$; ****$p < 0.0001$; ns−difference not significant).

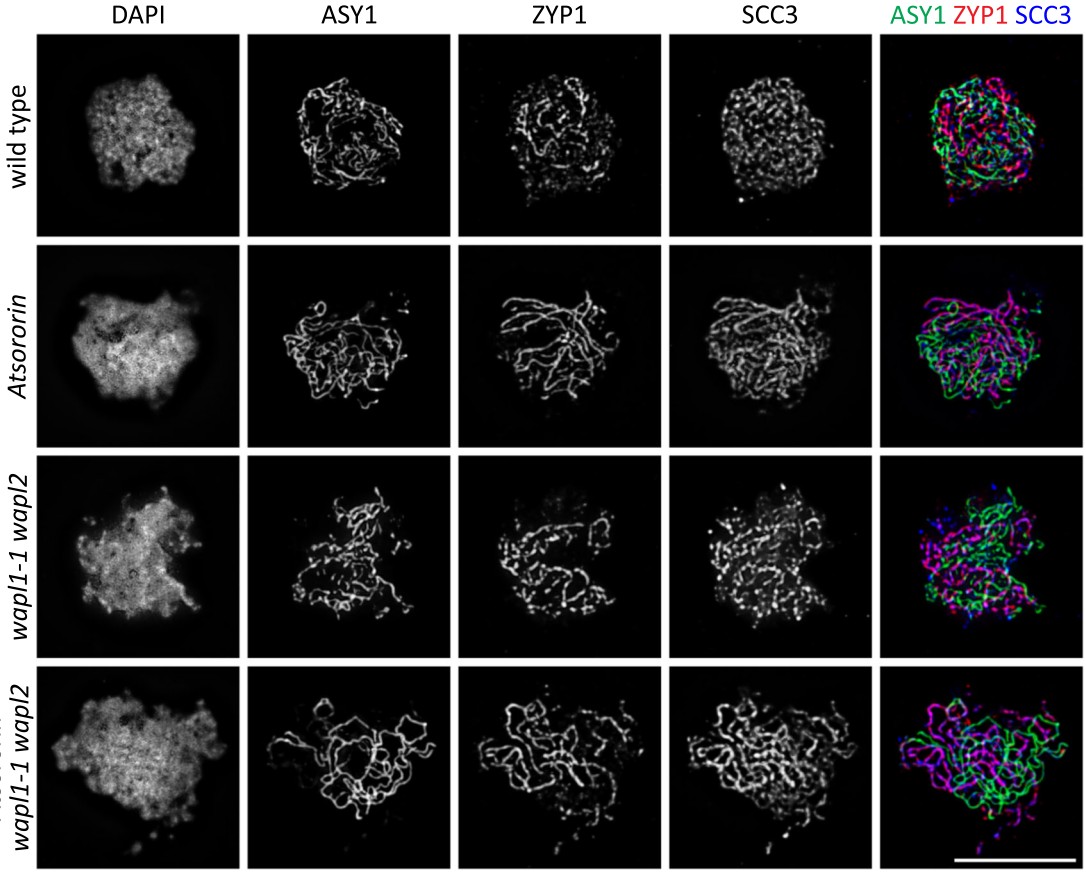

**Fig. 7 | Immunolocalization of the axis protein ASY1, the synaptonemal complex protein ZYP1 and the cohesin subunit SCC3 in male meiocytes during late zygotene in wild type plants and *Atsororin*, *wapl1-1 wapl2* and *Atsororin wapl1-1 wapl2* mutants.** Absence of AtSORORIN does not influence their time of deposition or their relative localization on meiotic chromosomes. For each genotype at least 3 pictures, all showing similar results, were acquired. Scale bar = 10 μm.

We confined the HMM-model to a region with highest conservation (*S. pombe* SPAC9E9.05.1: 298-311), using only fungi proteins, and searched specifically in *Saccharomycetes* species. In *Lipomyces starkeyi*, the best hit was significant (jgi|Lipst1_1|72111|Locus1483v3rpkm29.51, e-value 0.0041, score 20.9) and located at the c-terminus as well. Similarly, in the *Yarrowia lipolytica* proteome we selected YALI0C19756p (e-value 0.03, score 17.7). However, no candidate could be identified in *Saccharomyces cerevisiae* or in *Candida species*.

To identify plant candidates, we used the same HMM model as for the *S. pombe* screen before and searched within the *Arabidopsis thaliana* proteome. The best hit was to an unknown protein (AT3G56250.1, e-value 0.04, score 14.7), which is a member of a plant-specific protein family. Like for the Sororin family and the fungi candidates, the highest conservation lies in the C-terminal region. Except for some plant species, such as *Oryza sativa Japonica*, only one candidate gene was identified per genome.

The proteomes used in this study were retrieved from the NCBI-protein database (http://www.ncbi.nlm.nih.gov/protein) besides for *Saccharomyces cerevisiae* (http://www.yeastgenome.org/), *Schizosaccharomyces pombe* (http://www.pombase.org/), *Lipomyces starkeyi* (http://genome.jgi.doe.gov/Lipst1_1) and *Arabidopsis thaliana* (http://www.arabidopsis.org/).

Mulitple alignments were performed with MAFFT (version 7, L-INS-I method)[82], secondary structure prediction with Jpred (v4)[83]; and analysed in Jalview[84].

For the taxonomic tree, Wapl homologues were identified in an HMM search within the NCBI non-redundant protein database[85] or UniProt reference proteomes[86] using an alignment with representative sequences, including human WAPL NCBI|XP_042946.3:730-1269 and budding yeast Rad61 UniProt|Q99359: 260-633 (HMMER suite version 3.3.2, E-value cutoff 1e−7). HASPIN proteins were assigned with NCBI blast searches in the NCBI non-redundant protein database or in

## Table 1 | *S. pombe* strains and genotypes

| Strain number | Genotype |
|---|---|
| MG1 | *cen2(D107):kan-ura4+-lacO his7+::lacI-GFP* |
| MG2 | *cen2(D107):kan-ura4+-lacO his7+::lacI-GFP sor1::kanMX* |
| MG3 | *cen2(D107):kan-ura4+-lacO his7+::lacI-GFP wpl1::natMX* |
| MG4 | *cen2(D107):kan-ura4+-lacO his7+::lacI-GFP sor1::kanMX wpl1::natMX* |
| MG5 | *cen2(D107):kan-ura4+-lacO his7+::lacI-GFP eso1-G799D* |
| MG6 | *cen2(D107):kan-ura4+-lacO his7+::lacI-GFP sor1::kanMX eso1-G799D* |
| MG7 | *cen2(D107):kan-ura4+-lacO his7+::lacI-GFP mis4-242* |
| MG8 | *cen2(D107):kan-ura4+-lacO his7+::lacI-GFP sor1::kanMX mis4-242* |
| MG9 | *sor1-Pk9::kanMX pds5-myc::kanMX* |
| MG10 | *pds5-myc::kanMX* |
| MG11 | *sor1-Pk9::kanMX* |
| MG12 | *sor1-Pk9::kanMX psm3-GFP::natMX* |
| MG13 | *sor1-D303A-Pk9::kanMX psm3-GFP::natMX* |
| MG14 | *sor1-GFP::kanMX* |
| MG15 | *sor1-Flag::kanMX* |
| JG17331 | *ade6-M216 lys1-37* |
| JG16900 | *eso1-G799D* |
| JG16897 | *sor1::kanMX eso1-G799D* |
| JG16904 | *eso1-G799D sor1::sor1-wt::hphMX* |
| JG16879 | *eso1-G799D sor1::sor1-F299A::hphMX* |
| JG16881 | *eso1-G799D sor1::sor1-V302A::hphMX* |
| JG16883 | *eso1-G799D sor1::sor1-D303A::hphMX* |
| JG16885 | *eso1-G799D sor1::sor1-Y305A::hphMX* |
| MG13 | *sor1::TAP::kanMX* |
| MG14 | *sor1-F299A::TAP::kanMX* |
| MG15 | *sor1-V302A::TAP::kanMX* |
| MG16 | *sor1-D303A::TAP::kanMX* |
| MG17 | *sor1-Y305A::TAP::kanMX* |

(other auxotrophic markers not scored).

UniProt reference proteomes, using a representative set of homologues (covering the region corresponding to human HASPIN UniProt| Q8TF76:376-792) and applying highly restrictive E-value cutoffs (1e-10). The taxonomic tree was generated with the NCBI Taxonomy Common Tree[87]. The visualization was performed in iTOL v6[88].

### *S. pombe* methods

The genotypes of *S. pombe* strains used in this study are listed in Table 1. Standard YES media were used to grow *S. pombe* strains strains[89–91]. Tagging and deletion of *S. pombe* genes was performed according to our protocols described in ref. 92 and ref. 93, respectively. The immunofluorescence and microscopy techniques used to analyse chromosome segregation were performed as described in ref. 94. Point mutations in the *sor1* gene (to yield sor1-F299A, sor1-V302A, sor1-D303A and sor1-Y305A variants proteins) were introduced into the cloned *sor1* gene using the QuikChangeII kit (Agilent Technologies) and inserted into the genome by transformation.

For Western blot analyses, proteins were separated by electrophoresis through 12% polyacrylamide gels containing SDS (0.1%) and transferred to a PVDF membrane (Millipore). The membrane was blocked with 2% (w/v) milk-PBS-T (phosphate buffer saline buffer with 0.1% (v/v) Tween-20) and probed with antibodies. TAP-tagged proteins were detected using rabbit antiperoxidase antibody linked to peroxidase (PAP, Dako; 1:10,000 dilution). Tubulin was detected using mouse-anti-α-tubulin antibody (Sigma-Aldrich T5168; 1:10,000 dilution) and rabbit anti-mouse IgG-HRP secondary antibody (Santa Cruz

Biotechnology; 1:5000 dilution). GFP-tagged proteins were detected using mouse anti-GFP antibody (Roche 1814460, 1:1000 dilution) and anti-mouse-HRP antibody (Amersham, 1:5000). Pk-tagged proteins were detected using mouse-anti-Pk (V5) antibody (Serotec; 1:2000 dilution) and goat anti-mouse IgG-HRP secondary antibody (Santa Cruz Biotechnology; 1:5000 dilution) in 0.1% PBS-T. Myc-tagged proteins were detected using rabbit c-Myc antiserum (CM-100, Gramsch, Germany, 1:10,000 dilution) and secondary mouse anti-rabbit-IgG antibody conjugated to HRP (sc-2357, Santa Cruz Biotechnology, 1:20,000 dilution).

For coimmunoprecipitation, 10 ml of exponentially growing cells were collected, washed and lysed in 300 µL of IPP150 buffer [50 mM Tris-Cl (pH = 8.0), 150 mM NaCl, 10% glycerol, 0.1% NP-40, 1 mM PMSF and complete EDTA-free protease inhibitors] using glass beads as described in ref. 95. The lysates were centrifuged and subjected to affinity purification via binding to anti-GFP or anti-V5 agarose beads (Sigma-Aldrich) for 1 h at 4 °C. After washing with IPP150 buffer (3 × 1.5 ml), the bound proteins were released by the addition of SDS–PAGE sample buffer at 95 °C for 3 min. The presence of tagged proteins in the immunoprecipitates was detected by Western blot analysis as described above.

Nitrogen-starved cells were prepared as previously described[96]. Cells exponentially grown in EMM2 medium to a density of $10^7$ cells/ml at 30 °C were harvested by centrifugation and washed twice with EMM2-N (EMM2 lacking $NH_4Cl$) medium. Cells were resuspended in EMM2-N at a concentration of $10^7$ cells/ml and incubated at 30 °C for 21 h. Nitrogen starved cells were centrifuged and resuspended in EMM2 medium. Cells were harvested at 1, 2 and 3 h. Proteins were extracted from 50 ml cultures by trichloroacetic acid (TCA) precipitation according to Grallert and Hagan[97]. 50 µg of proteins were separated by electrophoresis through 4–12% polyacrylamide gels and transferred onto a PVDF membrane (Immobilon-P with pore size 0.45 µm; from Millipore). Anti-Pk antibody (mouse-anti-Pk (V5) antibody, Invitrogen R960-25, 1:2000 dilution) and goat anti-mouse IgG-HRP secondary antibody (Jackson ImmunoResearch 115-035-033, 1:10,000 dilution) were used to detect Sor1-Pk. Anti-tubulin antibody (Sigma-Aldrich T5168; 1:10,000 dilution) and rabbit anti-mouse IgG-HRP secondary antibody (Jackson ImmunoResearch 115-035-033, 1:10,000 dilution) were used to detect alpha tubulin as a loading control. To measure the DNA content of the cells, flow cytometry was performed according to[98] using Sytox Green (S7020, Invitrogen) and FACS NovoCyte Penteon (Agilent).

### In vitro APC/C assay

In vitro-translation of Sor1-HA-His by using reticulocyte lysate was performed according to the manufacturer's protocols (TNT Quick Couled Transcription/Translation System, Promega). Xenopus CSF-arrested egg extracts were prepared as described[99]. For interphase extracts, 0.4 mM $CaCl_2$ was supplemented to CSF extracts at a final concentration of 0.4 mM to induce exit from meiosis.

### Plant mutant lines and growth conditions

The *Arabidopsis thaliana* Columbia (Col-0) ecotype was used as wild-type reference. *Atsororin* mutant plants were generated via CRISPR-Cas9 (see below). The *wapl1-1 wapl2* double mutant (SALK_108385, SALK_127445)[35] was crossed with heterozygous *AtSORORIN* +/− mutant to obtain the *Atsororin wapl1-1 wapl2* triple mutant. Plants were grown on soil or on media plates containing Murashige and Skoog agar medium[100] with 2% sucrose. Long day growth conditions were applied with cycles of 16 h light and 8 h dark, at 21 °C and 60% humidity.

Leaves from rosette-stage plants grown on soil or the first true leaves from seedlings grown on plates, were collected for DNA isolation and genotyping. Mutants were confirmed by PCR using the primers listed below (Table 2). *Atsororin* mutants were confirmed by Sanger Sequencing of the PCR product (Supplementary Fig. 3a).

**Table 2 | Primers used in this study**

| Name | Sequence (5′ → 3′) | Utility |
|---|---|---|
| 35Sp_Fwd | CACTGACGTAAGGGATGACGCAC | PCR for genotyping BASTA gene |
| Basta_Rev | GAAGTCCAGCTGCCAGAAAC | |
| WAPL1.1LP | TCCAATTTAGTGAAACGTGGG | PCR for genotyping *WAPL1-1* T-DNA insertion line |
| WAPL1.1RP | ACACACTTGATTGAGAACCCG | |
| WAPL2LP | TCCAGCAAAACAGACAGGAAG | PCR for genotyping *WAPL2* T-DNA insertion line |
| WAPL2RP | CTCAAATCTGCGAACGAAGAG | |
| LBb1.3 | ATTTTGCCGATTTCGGAAC | T-DNA border primer for T-DNA insertion lines genotyping |
| Sororin_geno_Fwd | ATTATCGTCTCAAGCTCTCTCG | PCR for amplifying *SORORIN* gene |
| Sororin_geno_Rev | GCAGACATACGGCGAGTTAC | |
| Sororin_sequencing | GCTCTCTCGAGCCTTCTTCA | Sanger sequencing of the PCR product of *SORORIN* gene |
| Sororin_compl_Fwd | TCGGTCCAAATATATCAACAGC | PCR for genomic AtSORORIN for complementation line |
| Sororin_compl_Rev | AAATCGCCACTTCTGTACGC | |
| Sor_qPCR_For | ACACGGTAAGAAGGAAGGCC | qPCR of *AtSORORIN* |
| Sor_qPCR_Rev | AAGCTGCACTAACCGGATCC | |
| qPCR-ACTIN7-fwd | TTGCTGACCGTATGAGCAAAGA | qPCR of *ACTIN7* reference gene |
| qPCR-ACTIN7-rev | TCGATGGACCTGACTCATCGT | |
| Sor_Fwd_delATG | TTTTTTCCATGGAAGCTCCTCGCTCCG | PCR for amplification of *AtSORORIN* cDNA |
| SOR_cDNA_Rev | TTTTTTGAATTCTTAGTCTGAGTCGCTATTAGATACCT | |
| M13_Rev_2 | GTGGAATTGTGAGCGGATAAC | Sanger sequencing of *AtSORORIN* cDNA cloned into pCR Blunt TOPOII |

## Floral dip transformation of *A. thaliana*

*Arabidopsis* was transformed via *Agrobacterium tumefaciens* mediated DNA transfer. In brief, an aliquot of *A. tumefaciens* electroporation-competent cells was thawed on ice and 100 ng of plasmid were added. After 15 min incubation on ice, the cells were transferred to electroporation cuvettes (Eppendorf, 4307-000-593). After electroporation (400 Ω, 25 µF, 2.5 kV), 900 µL of SOC media were added to the cuvettes and cells were left to rest for 1 h at RT. 300 µL of transformed cells were plated on 2xTY plates supplemented with 50 µg/ml gentamycin, 50 µg/ml rifampicin and 100 µg/ml kanamycin (plasmid selection). Plates were left at 30 °C overnight.

A single colony from transformed *Agrobacterium tumefaciens* was inoculated into 500 mL of 2xTY medium supplemented with antibiotics. After 2 days rotating at 30 °C, cells were centrifuged at 4500 × *g* for 30 min at 4 °C. The pellet was then resuspended in 200 mL infiltration buffer (5% sucrose in dH₂O). Another centrifugation at 4500 × *g* for 30 min at 4 °C was performed and cells were now resuspended in 200 mL infiltration buffer containing 40 µL Silwet-L77. Prior to dipping the plants into the solution, their already developed siliques and open flowers were removed. Plants were then dipped into the Agrobacterium/infiltration buffer solution for 30 seconds and wrapped into plastic bags afterwards to avoid fast drying of the bacterial solution. Plants were transferred to the growth chamber and two days later the bags were removed.

## *Atsororin* mutant generation

The *Atsororin* mutant was generated by using the CRISPR-CAS9 technology. The gRNA sequence 5′-CCGTCGGAGGAAGAATACAG-3′ is specific to exon 1 of the *ATSORORIN* gene (At3g56250) and induces cleavage a few nucleotides downstream of the ATG codon. The gRNA was cloned into pGGE000-EF_pChimera2, and together with the Cas9 promoter in pGGA000-AB_PcUbi, the Cas9 version in pGGB000-BC_PuCas9 and the Cas9 terminator in pGGC000-CD_PeaTer further subcloned into the destination vector pGGZ003 utilizing the GOLDENGATE technique. The final plasmid was used to transform *Col*-0 plants by using the floral dip method[101]. Transgenic plants grown on soil were identified and selected by their resistance to the herbicide Basta (applied by spraying 13.5 mg/l). For subsequent generations we screened for the absence/presence of the BASTA resistance gene (*PAT*)

using the primers 35Sp_Fwd and Basta_Rev. Offspring of the initial transformants with or without the transgene were analysed for the presence of a mutation in the first exon 1 of the *AtSORORIN* gene. To do so, PCR amplicons were generated using the primers Sororin_geno_Fwd and Sororin_geno_Rev and subsequently sequenced with the primer Sororin_sequencing (Table 2). Plants with a mutation signature were grown for one or two more generations to identify individuals that inherited the mutation. We finally obtained a line without transgene and a stable heterozygous mutation in the *AtSORORIN* gene (Fig. 1). The *Atsororin* mutant line contains a 5 bps deletion within the first exon, 25 nucleotides down-stream of the ATG start codon (Fig. 3a; Supplementary Fig. 3a). It results in a premature TAA stop codon after generating a short peptide of 18 amino acid residues.

## RNA extraction and qPCR

RNA was extracted from 30 mg of *Arabidopsis thaliana* wild-type Col-0 and *Atsororin* −/− buds, cauline leaves and rosette leaves with the SV Total RNA Isolation System (Promega, Z3100). For each condition and genotype three separate extractions were performed. RNA samples were quantified on a DS-11+ Series Spectrophotometer (DeNovix) and 100 ng of each sample was used for first-strand cDNA synthesis (iScript kit Bio-Rad, 1708890). qPCR was performed using KAPA SYBR® FAST (Sigma-Aldrich, KK4601) and a RealPlex MasterCycler (Eppendorf, 6302000.504) according to the master mix protocol. *AtSORORIN* expression was calculated using the ΔΔCt method[102] normalized to ACTIN7 (AT5G09810) and the relative expression was related to the sample with the lowest Ct value (used primers are listed in Table 2).

## Detection of mutant *Atsororin* mRNA

The specific *AtSORORIN* cDNA was generated using RNA extracted from cauline leaves of *Atsororin* −/− mutant plants. To this end, the primers Sor_Fwd_delATG and SOR_cDNA_Rev were used and the resulting cDNA amplicons were ligated into the pCR Blunt TOPO II cloning vector and subsequently sequenced using the primer M13_Rev_2 (all primers are listed in Table 2) (Supplementary Fig. 3a).

## Complementation of *Atsororin* mutation

For complementing the *Atsororin* mutation, we first amplified the wild type *AtSORORIN* genomic version of the gene by PCR using Phusion

DNA Polymerase. The primers specific for the amplification are listed in Table 2. The amplicon was then cloned into the pCB302 vector[103], which is compatible with *A. tumefaciens* transformation and contains the BASTA resistance gene for future plant selection. Heterozygote *AtSORORIN* +/− plants were transformed with the pCB302 vector containing the *AtSORORIN* gene by the floral dip method to obtain the T1 generation of transformant plants. Two-week-old plants were selected for positive transformants by spraying the herbicide BASTA (150 mg/L BASTA in H₂O). Heterozygote *AtSORORIN* +/− plants (based on sequencing) and BASTA-resistance were selected for three more generations. The offspring of several F3 plants were sown on soil to check their genotype. The analysed *Atsororin* complementation lines were those that only generated offspring containing the *Atsororin* mutant allele and the complementing transgene (parent plants were homozygous for both, the *Atsororin* mutant allele and the complementing transgene).

### Seed counts

Mature but still green siliques originating from the fifth to the thirtieth flower per stem were harvested into fixing solution (1 part of glacial acetic acid and 3 parts of 96% EtOH) for distaining. After one day, the solution was renewed and seeds inside siliques were counted manually under a binocular microscope.

### Alexander staining

For pollen viability assays, the anthers and pollens from mature flowers were dissected under the microscope. The individual anthers were placed on a slide and a few drops (~20 µl) of Alexander staining[104] buffer (500 µl of water, 250 µl of 87% glycerin, 100 µl of 96% Ethanol, 50 µl of 1% acid fuchsin, 10 µl of 1% malachite green, 5 µl of 1% orange G, 50 µl of glacial acetic acid) were added. The anthers were covered with a coverslip and the microscopic slide was then incubated at 50 °C overnight. Stained pollen grains were observed with a microscope equipped with a differential contrast interference microscopy optics. Viable pollen grains appear round, filled with red-stained cytoplasm and coated with a thin green layer, while non-viable pollen appear only green, often shriveled and lack red cytoplasm.

### Spreading of nuclei, fluorescence in situ hybridisation (FISH) and immunolocalization of meiotic proteins

For somatic cell preparations, the tissue of interest was fixed in Carnoy´s fixative (1 part of glacial acetic acid and 3 parts of 96% EtOH). After washings twice with TRIS buffer (10 mM TRIS pH 7.5, 10 mM EDTA, 100 mM NaCl), the plant material was disrupted with a plastic pestle in Lysis buffer (15 mM TRIS pH 7.5, 0.5 mM spermine, 2 mM EDTA, 80 mM KCl, 20 mM NaCl, 0.1% Triton X-100). The solution was then pipetted trough a 40-micron cell strainer and centrifuged at 500 × *g* for 3 min. The pellet was resuspended in 50 µL of lysis buffer and pipetted to a glass slide to let air-dry. In order to visualize meiotic progression, anther spreads were prepared as described in ref. 105.

Fluorescence in situ hybridization (FISH) was performed as described in ref. 106. In brief, slides with somatic or meiotic nuclei were washed twice with 2xSSC for 5 min. After 10 min in 4% paraformaldehyde, slides were quickly washed in water and transferred through an ethanol series (70%, 90% and 96% EtOH) and then left to dry. A Locked Nucleic Acid (LNA) probe was used to detect centromere regions (5′-TTGGCTACACCATGAAAGCTT-3′; Qiagen). 20 µL of probe mix (250 nM LNA probe, 10% dextran sulfate, 50% formamide in 2 x Saline Sodium Citrate) were pipetted on the slide. A coverslip was applied, and the slides were placed on a hot plate at 75 °C for 4 min. After an overnight incubation at 37 °C, the slides were washed twice in 2xSCC and 15 µL of 2 µg/ml 4′,6 diamidino-2-phenylindol (DAPI) diluted in Vectashield (Vector Laboratories) were applied.

Spreads of nuclei for the detection of meiotic chromatin and associated proteins were performed as previously described[107].

Primary antibodies were used as follows: 1:10,000 anti-ASY1 raised in guinea pig[106], 1:500 anti-ZYP1 raised in rat[108], 1:500 anti-SCC3 raised in rabbit[109] and 1:250 anti-REC8 raised in rabbit[110]. The secondary antibodies are all commercially available and were used as follows: anti-guinea pig conjugated to Alexa Fluor 488 (1:400), anti-rabbit conjugated to Alexa Fluor 568 (1:400) and anti-rat conjugated to Alexa Fluor 647 (1:200).

Images were obtained with a Zeiss Axioplan microscope (Zeiss, Oberkochen, Germany) using a Quantix® CCD camera (Photometrics, Tucson, USA). Picture acquisition was performed with MetaMorph® Micoscopy Automation & Image Analysis software (Molecular Devices, Sunnyvale, USA). For meiotic prophase nuclei, Z-stacks with 100 nm intervals were acquired. Deconvolution was performed using AutoQuant software (Media Cybernetics Inc, Rockville, USA) and projections were done using Helicon Focus software (HeliconSoft, Kharkov, Ukraine).

### DAPI immunostaining

For the detection of SCC3 in somatic cells of inflorescences of *Arabidopsis thaliana* wild-type *and Atsororin* −/− plants material was collected into Carnoy's fixative. DAPI immunostainings were prepared according to Chelysheva et al.[111]. Rabbit anti-SCC3 1:500[109] was incubated in humid chamber at 4 °C O/N, later the slides were washed in PBS-T and incubated with anti-rabbit conjugated to Alexa Fluor 568 (1:40) at 37 °C for 1 h. Slides were mounted with 2 µg/ml 4',6 diamidino-2-phenylindol (DAPI) diluted in Vectashield (Vector Laboratories) and images were obtained with a Zeiss Axioplan microscope (Zeiss, Oberkochen, Germany) using a Quantix® CCD camera (Photometrics, Tucson, USA).

### Root tip image processing

Whole roots from 2-week-old plants grown on plates were collected from different genotypes and immersed in a solution of 10 µg/mL DAPI with 0.1% Triton-X100. After 30 min incubation at room temperature, roots were placed on a slide.

Imaging was performed with a Zeiss LSM710 microscope equipped with an AiryScan Unit. To generate the movies, Z-stacks with 250 nm intervals were acquired. Deconvolution was performed with the Huygens Software.

### Statistical analyses

All statistical analyses were performed using the GraphPad Prism 7 software. First, D'Agostino-Pearson omnibus normality test was performed to analyse if the data followed a Gaussian distribution. If yes, the two variables were compared using unpaired t-tests. When no Gaussian distribution was detected, unpaired Mann-Whitney tests were applied. Contingency tables were generated to compare expected (or wild type) data with mutant values. Fischer's exact test was used when two variables were compared. For three or more variables, Chi-square tests were performed.

### Reporting summary

Further information on research design is available in the Nature Portfolio Reporting Summary linked to this article.

## Data availability

All data supporting the findings of this study are available within the paper and its Supplementary Information. Source data are provided with this paper.

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

## Acknowledgements

J.G. was supported by the Austrian Science Fund (FWF) (grant P30516), the Slovak Grant Agency VEGA (1/0450/18 and 2/0026/18) and the Slovak Research and Development Agency (APVV-17-0130, APVV-18-0219, and APVV-16-0120). Research in the laboratory of P.S. was supported by the Austrian Science Fund (FWF) (I 1468-B16; Special Research Focus programme "Chromosome Dynamics" F3408-B19; FWF Doctoral Programme "Chromosome Dynamics" W1238-B20; Special Research Focus programme "Meiosis" F 8808-B) and the Austria's Agency for Education and Internationalization (Ernst Mach Grant to T.T.N.). Research in the laboratory of J.-M.P. is supported by Boehringer Ingelheim, the Austrian Life Sciences Programme 2023 (LS23 IF, project FO999902549), the European Research Council under the European Union's Horizon 2020 Research and Innovation Programme (1020558), the Human Frontier Science Programme (RGP0057/2018), and the Vienna Science and Technology Fund (LS19-029). J.-M.P. is also an adjunct professor at the Medical University of Vienna. We thank Vera Schoft and the Vienna Biocenter Core Facility (PlantS) for generating the *Atsororin* mutant line. We thank M. Yanagida, J.P. Javerzat and J. Gerton for sending yeast strains and Z. Benko, B. Huraiova, L. Cipak and S. Polakova for help with yeast experiments.

## Author contributions

A.S. and J.-M.P. performed the bioinformatic analyses and identified putative Sororin homologues in eukaryotes. M.G., I.K., T.N., G.L. and J.G. conceived and performed the experiments with *S. pombe*. I.P.M., C.F.S. and P.S. conceived and performed the experiments with *A. thaliana*. T.T.N. generated the *A. thaliana sororin* mutant. I.P.M., J.G., J.-M.P. and P.S. analysed the data and wrote the manuscript.

## Competing interests

The authors declare no competing interests.
