## [Peer Review File · Nature Communications]

Sororin is an evolutionary conserved antagonist of WAPLReviewer #1 (Remarks to the Author):

Sororin is an important cohesin regulator that maintains sister chromatid cohesion and acts as a WAPL antagonist. Sororin is conserved in vertebrates, but functional homologs have been found in other organisms such as drosophila. Orthologs are lacking in other organisms, such as yeast. It has been suggested that the yeast ortholog of sororin is the conserved acetyltransferase Eco1 (Zhang and Pati, 2012). The manuscript describes the identification of Sororin homologs in *S.pombe* and *A. thaliana* and explores their effect on cohesion in mitosis and meiosis. In *S. pombe*, Sor1 affects mitotic cohesin, while AtSororin in plants is implicated in chromosome segregation fidelity in vegetative growth and in meiosis. The interplay with WAPL, Eso1 and cohesin is studied. The paper provides an important and new insight into cohesin regulation in eukaryotes by addressing the question of how the evolutionarily conserved cohesin is regulated, in organisms that presumably lack sororin. However, some issues need to be further addressed to provide a comprehensive characterization of these new orthologs.

Comments:

In order to establish that Sor1 is a sororin ortholog in *S. pombe* its dynamics in yeast should be thoroughly analyzed and compared to vertebrates:

1. Sororin protein levels vary with the stages of the cell cycle peaking from S to G2.
2. In prophase, sororin dissociates from the arm and becomes cytosolic but remains on centromeres until metaphase.
3. Its not clear why Sor1 stability was tested in *Xenopus* extract and not *in vivo*.
4. The chromatin residency of Sor1 on chromatin and colocalization with cohesin should be determined by ChIP. The effect of Sor1 deletion of cohesin residency from *S. pombe* centromeres and chromosome arms should be tested.
5. Sororin is regulated by phosphorylation. It would be interesting to test if Sor1 is modified in a similar way.

The results suggest that sororin orthologs have a non-essential role in mitosis. The effect of AtSororin in plants suggests that it might be more important for meiosis. The impact on *S. pombe* meiosis should be examined.

The effect of overexpression should be tested in yeast and plants. Rankin et al., Molecular Cell 2005 showed that sororin overexpression results in failure to resolve and segregate sister chromatids in mitosis and an increase in the level of cohesin associated with metaphase chromosomes.

Despite the clear mitotic phenotype, the distribution of cohesin on mitotic chromosomes is similar in wild type and Atsororin mutant. Is this observation true also in mitotic cells?

Minor:

AtSororin expression should be determined. if antibodies are not available, the mRNA levels should be tested in a cell cycle-dependent manner. Validation for lack of mRNA in mutant plant cells should be presented.

The lacking orthologs in *S. cerevisiae* and other yeast is mentioned in the methods section. The results should be described in the main text, and ideally, data presented in a figure (an evolutionary tree or similar)

Discuss why normal cohesin levels are found in male meiocytes during late zygotene (Fig 7)

Reviewer #2 (Remarks to the Author):

The manuscript "SORORIN is an evolutionary conserved antagonist of WAPL" by Mota and co-authors presents the results of a range of experiments that provide convincing evidence that SORIN orthologs exist in *S. pombe* and *A. thaliana* and that while that SORORIN orthologs appear to share the common feature of antagonizing WAPL, there are subtle organismal and tissue specific

differences in the regulation of cohesion.

The results of the work are original and will make important contributions to the field of cohesion biology and chromatin structure in general. To my knowledge this is the first evidence that SORORIN exists in yeast and plants and the first direct in vivo evidence that SORORIN antagonizes WAPL. The authors also show that more than one WAPL antagonist exists in Arabidopsis and that these antagonists act in defined tissue and developmental contexts, most notably between vegetative and reproductive tissues and male and female meiosis and gametophyte development. Finally, the authors present evidence suggesting that AtSORORIN may have additional WAPL-independent functions in sister centromere protection at the male meiosis I to meiosis II transition, similar to the situation in *Drosophila*.

The manuscript is well-written, and the data presented is of high quality and supports the conclusions drawn. I only have a few small suggestions for improvement. 1) On page 9, line 243-251, the authors should clarify whether they are referring to Atsororin $-/-$ anthers or Atsororin $-/+$ anthers. As presented, this is confusing. If the images and data presented is for Atsororin $-/-$ mutant anthers, what do the Atsororin $-/+$ anthers look like. Based on the segregation data presented in Fig. 3G, it appears that Atsororin plants exhibit a drastic reduction in male fertility. 2) Pg 11, line 292, I believe that this should be 10.3 ± 0.5746 , not $10,3 \pm 0,5746$.

Reviewer comments and responses

We are grateful for all reviewer comments and are addressing all of them below. In addition, we up-dated the author list, corrected several spelling mistakes and added a few references. All changes to the originally submitted manuscript are highlighted in the revised manuscript file.

Reviewer #1 (Remarks to the Author):

Sororin is an important cohesin regulator that maintains sister chromatid cohesion and acts as a WAPL antagonist. Sororin is conserved in vertebrates, but functional homologs have been found in other organisms such as *Drosophila*. Orthologs are lacking in other organisms, such as yeast. It has been suggested that the yeast ortholog of Sororin is the conserved acetyltransferase Eco1 (Zhang and Pati, 2012).

Many thanks for the input. We have now added the statement and the corresponding reference to the manuscript (Discussion section, lines 450 – 452 of revised manuscript).

The manuscript describes the identification of Sororin homologs in *S. pombe* and *A. thaliana* and explores their effect on cohesion in mitosis and meiosis. In *S. pombe*, Sor1 affects mitotic cohesin, while AtSororin in plants is implicated in chromosome segregation fidelity in vegetative growth and in meiosis. The interplay with WAPL, Eso1 and cohesin is studied. The paper provides an important and new insight into cohesin regulation in eukaryotes by addressing the question of how the evolutionarily conserved cohesin is regulated, in organisms that presumably lack Sororin. However, some issues need to be further addressed to provide a comprehensive characterization of these new orthologs.

Comments:

In order to establish that Sor1 is a Sororin ortholog in *S. pombe* its dynamics in yeast should be thoroughly analyzed and compared to vertebrates:

1. Sororin protein levels vary with the stages of the cell cycle peaking from S to G2.

We have performed an additional experiment with synchronized *S. pombe* cultures, testing the abundance of Sor1 (Sor1-Pk) showing that Sor1-Pk protein levels are lower in nitrogen starved/G1 cells as compared to cycling cells which are mostly in G2 (Supplementary Figure 2c, including an extended figure legend; additional text in the “Results” section: lines 165 – 167 of revised manuscript).

2. In prophase, Sororin dissociates from the arm and becomes cytosolic but remains on centromeres until metaphase.

We established a chromatin-immunoprecipitation assay to quantify Sor1 levels on various centromeric and chromosome arm loci. However, Sor1-Pk enrichment was only slightly above the control (untagged) sample (see Figure X (A) below). This is probably due to

very low expression levels of the Sor1 protein (see Figure X (B) below) and makes it difficult to draw definitive conclusions from Sor1-Pk chromatin-immunoprecipitation experiments.

Moreover, in fission yeast, very little is known about Separase-independent removal of cohesin from chromosomes. As fission yeast cells enter mitosis, a small fraction of cohesin is released from chromosomes in a Separase cleavage-independent manner¹. However, it is not clear whether this process is related to the prophase pathway of cohesin removal in higher eukaryotes.

3. It is not clear why Sor1 stability was tested in *Xenopus* extract and not *in vivo*.

The reason was that other APC/C substrates are recognized across a wide range of species in *Xenopus* egg extracts, for example Securin/Pds5 from budding yeast² and Cut2 from fission yeast. Hiro Yamano used this assay for *S. pombe* substrates^{3,4}.

The new experiment mentioned above (presented in Supplementary figure 2c of the revised manuscript), indicates that the steady state levels of Sor1-Pk fluctuate during the cell cycle. Nevertheless, a well-controlled experiment measuring the half-life would require either a promoter-shut off or a pulse-chase experiment. Furthermore, we would also need to investigate potential transcriptional regulation and Sororin-Cohesin interactions, which in vertebrates depend on cell cycle regulated Cohesin acetylation. This could also be the case in fission yeast. While we agree, that investigating *S. pombe* Sor1 stability/regulation *in vivo* would be interesting, it appears beyond the scope of this first report for the reasons mentioned above.

4. The chromatin residency of Sor1 on chromatin and colocalization with cohesin should be determined by CHIP. The effect of Sor1 deletion of cohesin residency from *S. pombe* centromeres and chromosome arms should be tested.

To address whether Sor1 co-localizes with cohesin on chromatin, we analyzed chromatin localization of Sor1-Pk and Rad21-Pk by chromatin immunoprecipitation followed by real-time PCR at ten cohesin-rich and ten cohesin-poor loci (see answer above and Figure X below). We were able to reproduce previous observation that Rad21-Pk was enriched at so-called cohesin-rich loci¹. Sor1-Pk was detected at both cohesin-rich and cohesin-poor loci. However, the enrichment was only slightly above the control (untagged) sample. This is probably due to very low expression levels of the Sor1 protein and makes it difficult to draw definitive conclusions about the possible co-localization of Sor1 and cohesin on chromatin.

While we agree that analyzing the effect of the *sor1* deletion on chromatin-binding of cohesin in fission yeast would be very interesting, we believe it would not change the general conclusions of our manuscript and it also is beyond the scope of this initial characterization.

5. Sororin is regulated by phosphorylation. It would be interesting to test if Sor1 is modified in a similar way.

Indeed, *S. pombe* Sor1 phosphorylation has already been reported earlier and it is known to be phosphorylated on several serine and threonine residues (S3, S17, S37, S38, S55, S56, S59, S84, S85, S213, S219, T16, T18, T88)⁵⁻¹⁰ but the role of these phosphorylation events has not yet been studied. We agree that it would be very interesting to analyze the biological significance of Sor1 phosphorylation but believe that such an analysis is beyond the scope of this first report.

The results suggest that Sororin orthologs have a non-essential role in mitosis. The effect of AtSororin in plants suggests that it might be more important for meiosis. The impact on *S. pombe* meiosis should be examined.

Our preliminary data suggest that *sor1Δ* deletion has no major effect on chromosome segregation during meiosis but *sor1Δ* cells show increased frequency of lagging chromosomes during meiosis (see Figure Y below). Future experiments will assess the role of Sor1 in fission yeast meiosis more thoroughly.

The effect of overexpression should be tested in yeast and plants. Rankin et al., Molecular Cell 2005 showed that Sororin overexpression results in failure to resolve and segregate sister chromatids in mitosis and an increase in the level of cohesin associated with metaphase chromosomes.

The effect of overexpression of Sor1 on cellular functions has not been directly studied. Yet, it has been shown that cells expressing Sor1-YFP from a strong *nmt1* promoter are viable¹¹. This suggests that it should be possible to analyze cells overexpressing Sor1. However, we have not (yet) performed such studies. While we think that interpretation of results obtained from overexpression studies is often difficult, we will carefully consider these for our future analyses.

Regarding overexpression of AtSORORIN in plants we would like to point out that overexpression experiments in plants are not trivial and often not successful due to silencing effects, but in principle we agree with the reviewer that overexpression would be an interesting approach. Nevertheless, weighing the possible gain of information against the anticipated risk/effort we believe that the experiment is beyond the scope of this study, which focuses on establishing SORORIN as a conserved WAPL-antagonist found in two organisms.

Despite the clear meiotic phenotype, the distribution of cohesin on meiotic chromosomes is similar in wild type and Atsororin mutant. Is this observation true also in mitotic cells?

To answer this question, we performed an additional experiment. Staining for the cohesion complex protein SCC3 in somatic cells of inflorescences of wild-type and *Atsororin* mutants showed no difference in SCC3 abundance on chromosomes during prophase and interphase, indicating normal distribution and abundance of cohesin in these cells in *Atsororin* mutants (Supplementary Figure 4a of the revised manuscript, including an extended figure legend; lines 335 – 338 of the revised manuscript).

Minor:

AtSororin expression should be determined. If antibodies are not available, the mRNA levels should be tested in a cell cycle-dependent manner. Validation for lack of mRNA in mutant plant cells should be presented.

We performed additional experiments as requested. An anti-AtSORORIN antibody has been generated but unfortunately lacked specificity/sensitivity (data not shown). We analyzed *AtSORORIN* mRNA in wild-type and mutant plants and in different plant tissues. *AtSORORIN* mRNA had a similar abundance in all analyzed plant tissues (rosette leaves, cauline leaves and buds). Since the CRISPR/Cas9-mediated *Atsororin* mutation caused a 5 bp deletion we did not expect the mRNA to be turned over differently. Indeed, the mRNA levels of wild-type and mutant plants regarding *AtSORORIN* are comparable. Importantly, we show that the mRNA produced in homozygous *Atsororin* mutant plants also contains the 5bp deletion (Supplementary Figures 3a and 3b of the revised manuscript, including an extended figure legend; lines 239 – 244 of the revised manuscript). We also obtained and sorted nuclei according to their cell-cycle stage (G1 or G2) and performed qPCR experiments for *AtSORORIN* mRNA/cDNA but unfortunately failed to detect the transcript in most samples and therefore obtained inconclusive results (not shown).

The lacking orthologs in *S. cerevisiae* and other yeasts is mentioned in the methods section. The results should be described in the main text, and ideally, data presented in a figure (an evolutionary tree or similar).

We added an additional figure (Supplementary Figure 1; including a new figure legend), depicting a taxonomic tree to illustrate the occurrence/absence of Wapl, Sororin and Haspin in various organisms from distant lineages and refer to it in the results and discussion section (lines 153 – 157 and 460 – 462 of the revised manuscript).

Discuss why normal cohesin levels are found in male meiocytes during late zygotene (Fig 7).

We assume that only a sub-fraction of cohesin complexes is regulated by AtSORORIN during plant meiosis. Specifically in meiosis I, SWI1¹² antagonizes WAPL during meiotic prophase I and AtSORORIN possibly only during the metaphase I to anaphase I transition. The latter is possible not apparent in a *swi1* mutant as all cohesive cohesin has

been lost during prophase, leading to premature separation of sister chromatids. Importantly, in a *Atsororin* mutant, sister chromatid cohesion is maintained until metaphase I, yet centromeric fusion is lost at anaphase I onset leading to a peculiar co-segregation of separated sister chromatids to the same pole. As only a small fraction of centromeric cohesin complexes seem affected in the *Atsororin* mutant, this may not be sufficient for detecting differences in cytological preparations. In the future we will investigate the precise epistatic and functional relation of SWI1 and AtSORORIN. We have extended the discussion accordingly (lines 485 – 494 of the revised manuscript).

Reviewer #2 (Remarks to the Author):

The manuscript “SORORIN is an evolutionary conserved antagonist of WAPL” by Mota and co-authors presents the results of a range of experiments that provide convincing evidence that SORORIN orthologs exist in *S. pombe* and *A. thaliana* and that while that SORORIN orthologs appear to share the common feature of antagonizing WAPL, there are subtle organismal and tissue specific differences in the regulation of cohesion.

The results of the work are original and will make important contributions to the field of cohesion biology and chromatin structure in general. To my knowledge this is the first evidence that SORORIN exists in yeast and plants and the first direct *in vivo* evidence that SORORIN antagonizes WAPL. The authors also show that more than one WAPL antagonist exists in Arabidopsis and that these antagonists act in defined tissue and developmental contexts, most notably between vegetative and reproductive tissues and male and female meiosis and gametophyte development. Finally, the authors present evidence suggesting that AtSORORIN may have additional WAPL-independent functions in sister centromere protection at the male meiosis I to meiosis II transition, similar to the situation in *Drosophila*.

The manuscript is well-written, and the data presented is of high quality and supports the conclusions drawn. I only have a few small suggestions for improvement. 1) On page 9, line 243-251, the authors should clarify whether they are referring to *Atsororin* $-/-$ anthers or *Atsororin* $-/+$ anthers. As presented, this is confusing. If the images and data presented is for *Atsororin* $-/-$ mutant anthers, what do the *Atsororin* $-/+$ anthers look like. Based on the segregation data presented in Fig. 3G, it appears that *Atsororin* plants exhibit a drastic reduction in male fertility. 2) Pg 11, line 292, I believe that this should be 10.3 ± 0.5746 , not $10,3 \pm 0,5746$.

We have now clarified in the text and figure legends, that anthers of homozygous mutants (*Atsororin* $-/-$) are presented in Figure 3h (and further figures) and also added an additional image of anthers from heterozygous ($+/-$) plants (Supplementary Figure 3e), showing a large number of aborted pollen grains. The indicated numbers have been fixed (lines 265 – 269 of the revised manuscript).

References

1. Schmidt, C. K., Brookes, N. & Uhlmann, F. Conserved features of cohesin binding along fission yeast chromosomes. *Genome Biol.* **10**, (2009).
2. Cohen-Fix, O., Peters, J. M., Kirschner, M. W. & Koshland, D. Anaphase initiation in *Saccharomyces cerevisiae* is controlled by the APC-dependent degradation of the anaphase inhibitor Pds1p. *Genes Dev.* **10**, 3081–3093 (1996).
3. Yamano, H., Gannon, J. & Hunt, T. The role of proteolysis in cell cycle progression in *Schizosaccharomyces pombe*. *EMBO J.* **15**, 5268–5279 (1996).
4. Izawa, D., Goto, M., Yamashita, A., Yamano, H. & Yamamoto, M. Fission yeast Mes1p ensures the onset of meiosis II by blocking degradation of cyclin Cdc13p. *Nature* **434**, 529–533 (2005).
5. Koch, A., Krug, K., Pengelley, S., Macek, B. & Hauf, S. Mitotic substrates of the kinase aurora with roles in chromatin regulation identified through quantitative phosphoproteomics of fission yeast. *Sci. Signal.* **4**, (2011).
6. Carpy, A. *et al.* Absolute proteome and phosphoproteome dynamics during the cell cycle of *Schizosaccharomyces pombe* (fission yeast). *Mol. Cell. Proteomics* **13**, 1925–1936 (2014).
7. Swaffer, M. P., Jones, A. W., Flynn, H. R., Snijders, A. P. & Nurse, P. Quantitative Phosphoproteomics Reveals the Signaling Dynamics of Cell-Cycle Kinases in the Fission Yeast *Schizosaccharomyces pombe*. *Cell Rep.* **24**, 503–514 (2018).
8. Kettenbach, A. N. *et al.* Quantitative phosphoproteomics reveals pathways for coordination of cell growth and division by the conserved fission yeast kinase Pom1. *Mol. Cell. Proteomics* **14**, 1275–1287 (2015).
9. Tay, Y. D. *et al.* Fission Yeast NDR/LATS Kinase Orb6 Regulates Exocytosis via Phosphorylation of the Exocyst Complex. *Cell Rep.* **26**, 1654-1667.e7 (2019).
10. Halova, L. *et al.* A TOR (target of rapamycin) and nutritional phosphoproteome of fission yeast reveals novel targets in networks conserved in humans. *Open Biol.* **11**, (2021).
11. Matsuyama, A. *et al.* ORFeome cloning and global analysis of protein localization in the fission yeast *Schizosaccharomyces pombe*. *Nat. Biotechnol.* **24**, 841–847 (2006).
12. Yang, C. *et al.* SWITCH 1/DYAD is a WINGS APART-LIKE antagonist that maintains sister chromatid cohesion in meiosis. *Nat. Commun.* **10**, (2019).

A**B**
Figure X.**(A) Comparison of Sor1 and Rad21 chromatin binding.**

To address whether Sor1 co-localizes with cohesin on chromatin, we analyzed chromatin localization of Sor1-Pk and Rad21-Pk by chromatin immunoprecipitation followed by real-time PCR at ten cohesin-rich and ten cohesin-poor loci. We were able to reproduce previous observation that Rad21-Pk was enriched at so-called cohesin-rich loci¹. Sor1-Pk was detected at both cohesin-rich and cohesin-poor loci. However, the enrichment was only slightly above the control (untagged) sample. This is probably due to very low expression levels of the Sor1 protein and makes it difficult to draw definitive conclusions about the possible co-localization of Sor1 and cohesin on chromatin. Please note that log scale is used, therefore differences between Rad21-Pk on cohesin-rich and cohesin-poor loci are not easy to see.

(B) Sor1 protein is expressed at low levels.

To compare Sor1 and Rad21 protein levels, protein extracts (and their dilutions in case of Rad21) from indicated strains were analyzed by Western blotting. Tubulin was used as a loading control.

Meiosis (anaphase I)

Meiosis (anaphase II)

Figure Y.

***sor1Δ* cells show increased frequency of lagging chromosomes during meiosis.**

Wild type and *sor1Δ* cells carrying homozygous or heterozygous *cen2-GFP* were fixed and stained with antibodies against tubulin and GFP. Nuclei were visualized by Hoechst staining. Means +/- standard deviations are shown (* $p < 0.05$; ** $p < 0.01$; ns – not significant).

Reviewer #1 (Remarks to the Author):

The authors addressed most of the comments satisfactorily, except in a few points (e.g., ChIP), where an effort to address the concerns has been made without providing a conclusive result due to reasonable technical holdbacks. This is a high-quality paper, and I support its publication.